# Inference-Time Text-to-Video Alignment with Diffusion Latent Beam Search

**Yuta Oshima[1]    Masahiro Suzuki[1]    Yutaka Matsuo[1]    Hiroki Furuta[2]†**
[1]The University of Tokyo    [2]Google DeepMind
yuta.oshima@weblab.t.u-tokyo.ac.jp

## Abstract

The remarkable progress in text-to-video diffusion models enables the generation of photorealistic videos, although the content of these generated videos often includes unnatural movement or deformation, reverse playback, and motionless scenes. Recently, an alignment problem has attracted huge attention, where we steer the output of diffusion models based on some measure of the content's goodness. Because there is a large room for improvement of perceptual quality along the frame direction, we should address which metrics we should optimize and how we can optimize them in the video generation. In this paper, we propose *diffusion latent beam search* with *lookahead estimator*, which can select a better diffusion latent to maximize a given alignment reward at inference time. We then point out that improving perceptual video quality with respect to alignment to prompts requires *reward calibration* by weighting existing metrics. This is because when humans or vision language models evaluate outputs, many previous metrics to quantify the naturalness of video do not always correlate with the evaluation. We demonstrate that our method improves the perceptual quality evaluated on the calibrated reward, VLMs, and human assessment, without model parameter update, and outputs the best generation compared to greedy search and best-of-N sampling under much more efficient computational cost. The experiments highlight that our method is beneficial to many capable generative models, and provide a practical guideline: we should prioritize the inference-time compute allocation into enabling the lookahead estimator and increasing the search budget, rather than expanding the denoising steps. [1] [2]

## 1 Introduction

The remarkable progress in text-to-video diffusion models enables photorealistic, high-resolution video generation [1–4]. Many future applications are anticipated, such as creating novel games [5], movies [6], or simulators to control real-world robots [7]. However, the detailed contents of the generated video often include unnatural movement or deformation, reverse playback, and motionless scenes, which should not happen in the real world. For instance, simulating factual physics in the generated video is still challenging [8, 9]. Recently, it has attracted a lot of attention to steering the output of diffusion models based on reward evaluation, quantifying the goodness of the content, which is studied as an alignment problem [10, 11]. There is a large room for improvement of perceptual quality along the frame direction in the video, and to align models with our preferences, we should address which metrics to optimize and how to optimize them.

---

†Work done as an advisory role only.
[1]Website: https://sites.google.com/view/t2v-dlbs
[2]Code: https://github.com/shim0114/T2V-Diffusion-Search

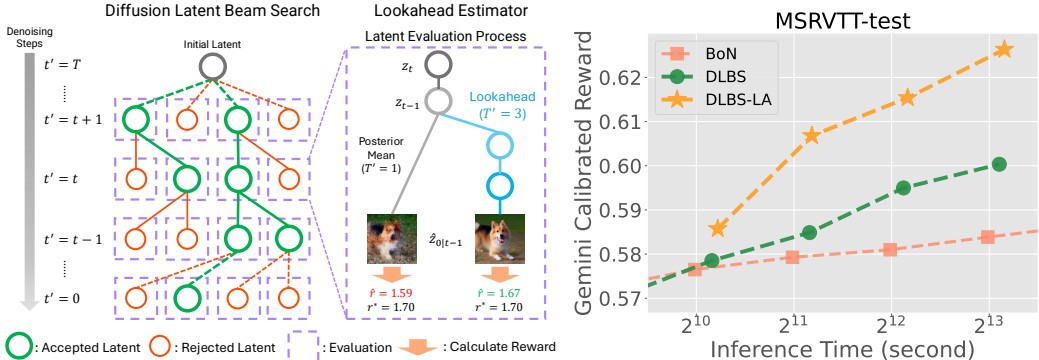

Figure 1: (**Left**) Diffusion latent beam search (DLBS) seeks a better diffusion path over the reverse process; sampling $K$ latents per beam and possessing $B$ beams for the next step, which mitigates the effect from inaccurate $\mathrm{argmax}$. Lookahead (LA) estimator notably reduces the noise at latent reward evaluation by interpolating the rest of the time steps from the current latent with deterministic DDIM. (**Right**) DLBS achieves much better computational-efficiency than best-of-N (BoN), as achieving higher performance gains under the same execution time. LA estimator (DLBS-LA) could remarkably boost efficiency only with marginal overhead on top of DLBS.

In this paper, we propose *Diffusion Latent Beam Search* (DLBS) with *lookahead estimator*, an inference-time search over the reverse process (Figure 1; **Left**), which can select a better diffusion latent to maximize a given alignment reward. A lookahead estimator reduces the noise in the reward estimate, and a beam search robustly explores the latent paths, avoiding inaccurate $\mathrm{argmax}$ operations.

We then point out that the improvement of perceptual video quality, considering the alignment to prompts, requires *reward calibration* of existing metrics [12]. When evaluating outputs using capable vision language models [13, 14] or human raters, many previous metrics for quantifying video naturalness do not always correlate with them. Optimal reward design for measuring perceptual quality highly depends on the degree of dynamics described in evaluation prompts. We design a weighted linear combination of multiple metrics, which is calibrated to perceptual quality and improves the correlation with VLM/human preference.

We demonstrate that DLBS can induce high-quality outputs based on the calibrated reward, AI, and human feedback (Figure 2), without model parameter update, and realize the best generation under much more efficient computational cost compared to greedy search [15, 11] and best-of-N sampling [16, 17]. The experiments also highlight that our method is beneficial to many SoTA models (e.g., Latte [18], CogVideoX [19], and Wan 2.1 [20]), and provide a practical guideline that we should prioritize the inference-time compute allocation into enabling the lookahead estimator and increasing the search budget, rather than expanding the denoising steps.

## 2 Preliminaries

**Latent Diffusion Models**  Latent diffusion models [21, 18] are a special class of diffusion probabilistic models [22, 23], and popular choices for high-resolution text-to-video generation [24, 25, 4], which considers the diffusion process in embedding space. Let $\mathbf{x}_0$ be a video and encode it as $\mathbf{z}_0 = \mathrm{Enc}(\mathbf{x}_0)$ using VAE [26]. Continuous-time forward diffusion process can be modeled as a solution to a stochastic differential equation (SDE) [27]: $d\mathbf{z} = \mathbf{f}(\mathbf{z}, t)dt + g(t)d\mathbf{w}$, where $\mathbf{z}_0 \sim p_0(\mathbf{z})$ is the latent as initial condition while $p_t(\mathbf{z})$ is the marginal distribution of $\mathbf{z}_t$, $\mathbf{f} : \mathbb{R}^d \times \mathbb{R} \to \mathbb{R}^d$ is the drift coefficient, $g : \mathbb{R} \to \mathbb{R}$ is the diffusion coefficient, and $\mathbf{w} \in \mathbb{R}^d$ is $d$-dimensional standard Wiener process. $\mathbf{f}(\cdot, \cdot)$ and $g(\cdot)$ are designed appropriately for the marginal distribution to reach $p_T(\mathbf{z}) \approx \mathcal{N}(0, \mathbf{I})$ as $t \to T$ [28]. Reverse diffusion process generates samples $\mathbf{z}_0$ through the following reverse-time SDE: $d\mathbf{z} = [\mathbf{f}(\mathbf{z}, t) - g(t)^2 \nabla_{\mathbf{z}} \log p_t(\mathbf{z})]dt + g(t)d\bar{\mathbf{w}}$, where $dt$ here is an infinitesimal negative time step from $T$ to 0 and $\bar{\mathbf{w}} \in \mathbb{R}^d$ is a standard reverse-time Wiener process. We start this with $\mathbf{z}_T \sim \mathcal{N}(0, \mathbf{I})$. This SDE induces the marginal distribution on the data $p^{\mathrm{pre}}(\mathbf{z})$ (i.e., pre-trained diffusion models). While we omit the notation for simplicity, we consider the text-to-video generation problem, where the diffusion process is conditioned on text prompts $\mathbf{c}$.

**Alignment for Text-to-Video Diffusion Models**  In this paper, we define the alignment problem in text-to-video generation as increasing the probability of generating perceptually good video for

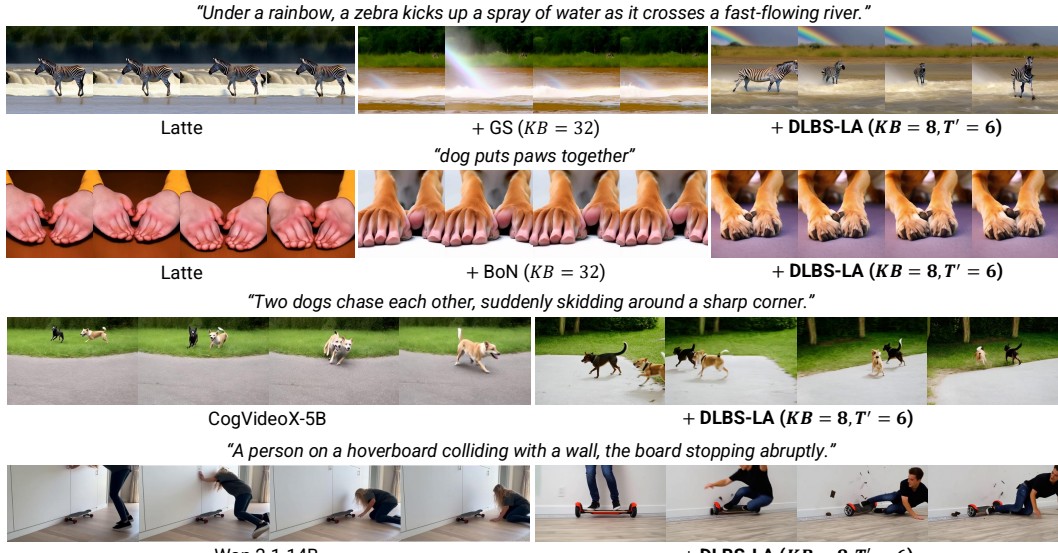

*"Under a rainbow, a zebra kicks up a spray of water as it crosses a fast-flowing river."*

Latte  +  GS ($KB = 32$)  + **DLBS-LA** ($KB = 8, T' = 6$)

*"dog puts paws together"*

Latte  + BoN ($KB = 32$)  + **DLBS-LA** ($KB = 8, T' = 6$)

*"Two dogs chase each other, suddenly skidding around a sharp corner."*

CogVideoX-5B  + **DLBS-LA** ($KB = 8, T' = 6$)

*"A person on a hoverboard colliding with a wall, the board stopping abruptly."*

Wan 2.1-14B  + **DLBS-LA** ($KB = 8, T' = 6$)

Figure 2: Comparison of text-to-video results between DLBS-LA, base models, and other sampling methods on SoTA models (Latte [18], CogVideoX [19], and Wan 2.1 [20]). DLBS-LA produces more dynamic, natural, and prompt-aligned videos than all baselines.

humans, such as $\max \mathbb{E}[p(\mathcal{O} = 1|\mathbf{x}_0, \mathbf{c})]$ where $\mathcal{O} \in \{0, 1\}$ represents if the generated video $\mathbf{x}_0$ conditioned on $\mathbf{c}$ is perceptually higher quality or not. The common assumption is such a probability depends on a proxy scalar reward function $r(\mathbf{x}_0, \mathbf{c})$ such as $p(\mathcal{O} = 1|\mathbf{x}_0, \mathbf{c}) \propto \exp(\beta^{-1} r(\mathbf{x}_0, \mathbf{c}))$ with $\beta \in \mathbb{R}$, and then the problem comes down to reward maximization. The proxy reward function may input the generated video $\mathbf{x}_0$ and a prompt $\mathbf{c}$.

**Alignment as Stochastic Optimal Control**  Previous works formulate such a reward maximization problem from the view of stochastic optimal control [29, 11, 30], where we aim to find an additional drift term $\mathbf{u}(\cdot, \cdot)$ for the following reverse SDE: $d\mathbf{z} = [\mathbf{f}(\mathbf{z}, t) - g(t)^2 \nabla_\mathbf{z} \log p_t(\mathbf{z}) + \mathbf{u}(\mathbf{z}, t)]dt + g(t)d\bar{\mathbf{w}}$. For convenience, we adopt the change-of-variables as $\boldsymbol{\nu}_t := \mathbf{z}_{T-t}$ and $\bar{\mathbf{f}}(\boldsymbol{\nu}, t) := \mathbf{f}(\boldsymbol{\nu}, t) - g(t)^2 \nabla_\boldsymbol{\nu} \log p_t(\boldsymbol{\nu})$ because stochastic control is often based on the standard flow of time ($t : 0 \to T$), and then the original SDE is re-written as: $d\boldsymbol{\nu} = [\bar{\mathbf{f}}(\boldsymbol{\nu}, t) + \mathbf{u}(\boldsymbol{\nu}, t)]dt + g(t)d\mathbf{w}$, where $dt$ here is an infinitesimal time step and $d\mathbf{w}$ is a standard Wiener process.

Because the alignment problem comes down to reward maximization, the objective in stochastic control literature is

$$\mathbf{u}^* = \underset{\mathbf{u}}{\arg\max} \, \mathbb{E}\left[ r'(\boldsymbol{\nu}_T) - \frac{\lambda}{2} \int_{t=0}^{T} \frac{\|\mathbf{u}(\boldsymbol{\nu}_t, t)\|^2}{g(t)^2} dt \right] \tag{1}$$

where $r'(\cdot) := r(\text{Dec}(\cdot))$ evaluates the latent in the video space and $\lambda > 0$. $\mathbb{E}[\cdot]$ is taken over sampling process above. In stochastic control, the optimal value function is known to be defined as,

$$v_t^*(\boldsymbol{\nu}) = \mathbb{E}_{p^*}\left[ r'(\boldsymbol{\nu}_T) - \frac{\lambda}{2} \int_{s=t}^{T} \frac{\|\mathbf{u}(\boldsymbol{\nu}_s, s)\|^2}{g(s)^2} ds | \boldsymbol{\nu}_t = \boldsymbol{\nu} \right], \tag{2}$$

where $p_t^*(\boldsymbol{\nu}) = \frac{1}{Z} \exp(\frac{v_t^*(\boldsymbol{\nu})}{\lambda}) p_t^{\text{pre}}(\boldsymbol{\nu})$, and obtain the optimal drift $\mathbf{u}^*(\boldsymbol{\nu}, t) = g(t)^2 \nabla_\boldsymbol{\nu} \frac{v_t^*(\boldsymbol{\nu})}{\lambda}$ [31]. This optimal value function is the solution of stochastic Hamilton-Jacobi-Bellman equation [32] according to this Feynman-Kac formula [33, 34]:

$$\exp\left( \frac{v_t^*(\boldsymbol{\nu})}{\lambda} \right) = \mathbb{E}_{p^{\text{pre}}}\left[ \exp\left( \frac{r'(\boldsymbol{\nu}_T)}{\lambda} \right) | \boldsymbol{\nu}_t = \boldsymbol{\nu} \right] \tag{3}$$

and then we obtain a tractable form of the optimal drift term as:

$$\mathbf{u}^*(\boldsymbol{\nu}_t, t) = g(t)^2 \nabla_\boldsymbol{\nu} \log \mathbb{E}_{p^{\text{pre}}}\left[ \exp\left( \frac{r'(\boldsymbol{\nu}_T)}{\lambda} \right) | \boldsymbol{\nu}_t = \boldsymbol{\nu} \right]. \tag{4}$$

The intuition here is that the optimal drift pulls the current latent $\boldsymbol{\nu}$, while following the pre-trained reverse SDE, into the region achieving a higher reward at time $T$.

# 3 Diffusion Latent Beam Search

We first provide a unified view of existing inference-time alignment methods through several practical approximations of optimal drift $\mathbf{u}^*(\boldsymbol{\nu}_t, t)$ (Section 3.1). To mitigate errors from approximations, we propose a novel search algorithm, *diffusion latent beam search* with *lookahead estimator* (Section 3.2).

## 3.1 A Unified View on Practical Approximations

While Equation 4 has a relatively tractable form, it is still computationally expensive, since the expectation requires complete diffusion sampling to evaluate the latent at each time step and face numerical instability. Previous alignment methods rely on multiple-step practical approximations.

**Step. 1: Jensen's Inequality** First, when assuming $\frac{r'(\boldsymbol{\nu}_T)}{\lambda}$ is almost deterministic (this might hold when $t \to T$), Jensen's inequality yields the following approximation by exchanging log and $\mathbb{E}[\cdot]$, which can be considered as a certain form of classifier guidance [35]:

$$\mathbf{u}^*(\boldsymbol{\nu}_t, t) \approx \frac{g(t)^2}{\lambda} \nabla_{\boldsymbol{\nu}} \mathbb{E}_{p^{\text{pre}}} \left[ r'(\boldsymbol{\nu}_T) | \boldsymbol{\nu}_t = \boldsymbol{\nu} \right]. \tag{5}$$

**Step. 2: Tweedie's Formula** To avoid the computationally expensive expectation, the expected reward is further approximated as $\mathbb{E}_{p^{\text{pre}}}[r'(\boldsymbol{\nu}_T) | \boldsymbol{\nu}_t = \boldsymbol{\nu}] \approx r'(\hat{\boldsymbol{\nu}}_{T|t})$ where $\hat{\boldsymbol{\nu}}_{T|t} \approx \mathbb{E}_{p^{\text{pre}}}[\boldsymbol{\nu}_T | \boldsymbol{\nu}_t = \boldsymbol{\nu}]$ is a one-step approximation of posterior mean [36], which can be calculated only with the current latent $\boldsymbol{\nu}_t$ without full diffusion path. Therefore, the optimal drift term can be seen as solely depending on the current time step $t$:

$$\mathbf{u}^*(\boldsymbol{\nu}_t, t) \approx \frac{g(t)^2}{\lambda} \nabla_{\boldsymbol{\nu}} r'(\hat{\boldsymbol{\nu}}_{T|t}). \tag{6}$$

Such a computationally tractable drift term has been leveraged for previous inference-time alignment methods via approximate guidance or twisted sequential Monte Carlo (SMC) [37]. However, as the approximated posterior mean $\hat{\boldsymbol{\nu}}_{T|t}$ in intermediate steps is noisy, evaluation with the reward function for clean data $r'(\cdot)$ may not pro-

---

**Algorithm 1** Diffusion Latent Beam Search (DLBS) with Stochastic DDIM

**Input:** latent diffusion model $\epsilon_\theta$, reward function $r'$, noise scheduling parameter $\{\alpha_t\}_{t=0}^T, \{\sigma_t\}_{t=0}^T$, number of beams $B$, number of candidates $K$
1: $\mathbf{z}_T^1, \cdots, \mathbf{z}_T^B \sim \mathcal{N}(\mathbf{0}, \mathbf{I})$  ▷ Initial $B$ beams
2: **for** $t = T$ to $1$ **do**
3:    **for** $j = 1$ **to** $B$ **do**
4:       ▷ Compute the posterior mean of $\mathbf{z}_{t-1}^j$
5:       $\hat{\mathbf{z}}_{0|t}^j = \frac{1}{\sqrt{\alpha_t}}(\mathbf{z}_t^j - \sqrt{1 - \alpha_t}\epsilon_\theta(\mathbf{z}_t^j))$
6:       $\mathbf{z}_{t-1}^j = \sqrt{\alpha_{t-1}}\hat{\mathbf{z}}_{0|t}^j + \sqrt{1 - \alpha_{t-1} - \sigma_t^2}\epsilon_\theta(\mathbf{z}_t^j)$
7:    **end for**
8:    **if** $t > 1$ **then**
9:       **for** $j = 1$ **to** $B$ **do**
10:         ▷ Sample $K$ next candidate latents
11:         $\mathbf{z}_{t-1}^{ij} = \mathbf{z}_{t-1}^j + \sigma_t \epsilon_t^i$ with $\epsilon_t^1, ..., \epsilon_t^K \sim \mathcal{N}(\mathbf{0}, \mathbf{I})$
12:         ▷ Estimate the clean sample from noisy latent
13:         $\hat{\mathbf{z}}_{0|t-1}^{ij} = \frac{1}{\sqrt{\alpha_{t-1}}}(\mathbf{z}_{t-1}^{ij} - \sqrt{1 - \alpha_{t-1}}\epsilon_\theta(\mathbf{z}_{t-1}^{ij}))$
14:      **end for**
15:      ▷ Search $B$ higher-reward beams from $KB$ latents
16:      $\texttt{budget} := \{(\mathbf{z}_{t-1}^{11}, \hat{\mathbf{z}}_{0|t-1}^{11}), \cdots, (\mathbf{z}_{t-1}^{KB}, \hat{\mathbf{z}}_{0|t-1}^{KB})\}$
17:      **for** $j' = 1$ **to** $B$ **do**
18:         $\mathbf{z}_{t-1}^{j'} = \text{argmax}_{\mathbf{z}_{t-1}^{ij} \in \texttt{budget}} \ r'(\hat{\mathbf{z}}_{0|t-1}^{ij})$
19:         $\texttt{budget} = \texttt{budget} \setminus \{(\mathbf{z}_{t-1}^{j'}, \hat{\mathbf{z}}_{0|t-1}^{\text{argmax}})\}$
20:      **end for**
21:      $j \in \{1, \cdots, B\} \leftarrow j'$  ▷ Reset selected $B$ indices
22:    **end if**
23: **end for**
24: **return:** $\mathbf{z}_0 = \text{argmax}_{\mathbf{z}_0^j \in \{\mathbf{z}_0^1, \cdots, \mathbf{z}_0^B\}} \ r'(\mathbf{z}_0^j)$

---

vide a reliable signal [38]. Moreover, Equation 6 requires the reward gradient, which is not applicable to non-differentiable rewards, such as AI feedback, and is also not suitable for modalities whose reward gradient imposes a huge computational cost in practice, such as video.

**Step. 3: Converting Reward Gradient into argmax** The usage of reward gradient can be converted into argmax operator [11, 39, 40]. The intuition here is that since the optimal drift in Equation 6 induces the diffusion latent to the direction where it maximizes the reward, we replace such a maximization with a zeroth-order search. The SDE is approximated as:

$$d\boldsymbol{\nu} = \bar{\mathbf{f}}(\boldsymbol{\nu}, t)dt + g(t)d\mathbf{w}^* \ \text{ where } \ d\mathbf{w}^* = \text{argmax}_{d\mathbf{w}} \ r'(\hat{\boldsymbol{\nu}}_{T|t}). \tag{7}$$

Note that the current diffusion latents $\boldsymbol{\nu}_t$ and posterior mean $\hat{\boldsymbol{\nu}}_{T|t}$ are sampled by following the standard Wiener process $d\mathbf{w}$. This approximation is leveraged for inference-time alignment via greedy search [11, 39] or SMC [41] of diffusion latents. However, greedy search can result in sub-optimal generation affected by inaccurate reward estimate $r'(\hat{\boldsymbol{\nu}}_{T|t})$ due to its noisy input. Moreover, it can be challenging to obtain an accurate density ratio term required in SMC for a high-dimensional domain, such as video generation.

## 3.2 Mitigating Approximation Errors via Beam Search

Existing practical algorithms based on these three approximations, such as greedy search [11, 39], fall into sub-optimal generation due to the erroneous reward evaluation with a noisy estimate of the posterior mean [36], and argmax operator based on them. To resolve the error accumulation, we propose a simple yet robust modification, *diffusion latent beam search (DLBS)* with *lookahead estimator*. To clearly describe the practical implementation, we use the notation of a discrete-time diffusion process in the rest of the section (see Appendix E for the continuous-time diffusion process).

**Practical Implementation**  We summarize the detailed sampling procedure of DLBS in Algorithm 1. For the diffusion sampler, we use stochastic DDIM [42] with a decreasing sequence $\{\alpha_t\}_{t=1}^{T} \in (0,1]^T$, noise level $\eta$, and noise schedule $\sigma_t = \eta\sqrt{(1-\alpha_{t-1})/(1-\alpha_t)}\sqrt{1-\alpha_{t-1}/\alpha_t}$, which is equivalent to DDPM [23] when $\eta = 1.0$. We initialize $B$ latent beams from the Gaussian distribution (Line 1.1), sample $K$ latents per beam in the next time step (Line 1.11), and then compute the one-step estimation of the posterior mean (Line 1.13). DLBS evaluates the estimator of posterior mean $\hat{\mathbf{z}}_{0|t-1}$ with reward function (Line 1.18) and selects Top-$B$-rewarded latent beams instead of Top-1 (i.e., argmax) from $KB$ candidates (Line 1.19), which is iterated over entire reverse process from $t = T$ to $t = 0$. DLBS can possess latent beams more widely than greedy search under the same budget, which mitigates error propagation due to the approximated diffusion latent evaluation.

**Lookahead Estimator**  The other source of approximation errors than the argmax operator is a one-step estimator of the posterior mean $\hat{\mathbf{z}}_{0|t-1}$ from Tweedie's formula, which is still noisy, especially in earlier time steps, and leads to inaccurate reward evaluation. To reduce errors in reward evaluation, we propose a lookahead (LA) estimator $\tilde{\mathbf{z}}_{0|\tilde{t}(0)}$, which is estimated by running $T'$-step deterministic DDIM $(1 < T' << T)$ while equally interpolating the rest of time steps from the current latent $\mathbf{z}_{t-1}$ to $\mathbf{z}_0$ (Algorithm 2). While requiring additional denoising steps, its cost is almost the same as naive DLBS because most computational costs come from when we decode $\mathbf{z}_0$ (i.e., reward evaluation). Theoretically, enlarging the lookahead steps $T'$ monotonically

---

**Algorithm 2** Lookahead (LA) with Deterministic DDIM

**Input:** latent diffusion model $\epsilon_\theta$, current diffusion latent $\mathbf{z}_{t-1}$, number of lookahead steps $T'(<< T)$

1: ▷ Run $T'$-step deterministic DDIM starting from $\mathbf{z}_{t-1}$
2: $\tilde{t}(s) \in \{t-1, \ldots, \lfloor \frac{s}{T'}(t-1)\rfloor, \ldots, \lfloor \frac{1}{T'}(t-1)\rfloor, 0\}$
3: Select new lookahead noise schedule $\{\tilde{\alpha}_s\}_{s=0}^{T'}$ for $T'$-**step interpolation** of the rest of original $\{\alpha_{t'}\}_{t'=0}^{t-1}$
4: $\mathbf{z}_{\tilde{t}(T')} := \mathbf{z}_{t-1}$
5: $\tilde{\mathbf{z}}_{0|\tilde{t}(T')} = \frac{\mathbf{z}_{\tilde{t}(T')} - \sqrt{1-\tilde{\alpha}_{T'}}\epsilon_\theta(\mathbf{z}_{\tilde{t}(T')})}{\sqrt{\tilde{\alpha}_{T'}}}$
6: **for** $s = T'$ to $1$ **do**
7:     $\mathbf{z}_{\tilde{t}(s-1)} = \sqrt{\tilde{\alpha}_{s-1}}\tilde{\mathbf{z}}_{0|\tilde{t}(s)} + \sqrt{1-\tilde{\alpha}_{s-1}}\epsilon_\theta(\mathbf{z}_{\tilde{t}(s)})$
8:     $\tilde{\mathbf{z}}_{0|\tilde{t}(s-1)} = \frac{\mathbf{z}_{\tilde{t}(s-1)} - \sqrt{1-\tilde{\alpha}_{s-1}}\epsilon_\theta(\mathbf{z}_{\tilde{t}(s-1)})}{\sqrt{\tilde{\alpha}_{s-1}}}$
9: **end for**
10: **return:** $(\mathbf{z}_{t-1}, \tilde{\mathbf{z}}_{0|\tilde{t}(0)})$ ▷ Latent and LA estimator

---

tightens the upper bound on the reward-approximation error (see Appendix F). Empirically, a modest horizon ($T' = 2, 3, 6$) delivers substantial search improvements (Figure 9; **Left**), but the marginal gains saturate, so pushing $T'$ further yields little additional benefit (see Appendix M.4).

## 4 Calibrating Reward to Preference Feedback

Human evaluation is one of the most valuable assessments for generative models, yet gathering human feedback at scale is prohibitively costly. A practical approach to reduce the time and cost is to leverage AI feedback from VLMs [43], which has been shown to modestly align with human judgment on video quality [44–46] (see Appendix K). In this work, we assume that the VLM evaluation works as an oracle, and we align model outputs with the preferences of VLMs, which is reasonable due to their capability and the cost to be saved. Our qualitative and quantitative evaluations also confirm that the highly rated video by VLMs is generally good for us.

However, because alignment via inference-time search requires massive reward evaluation queries, we still need to build more tractable proxy rewards that do not rely on humans or external VLM APIs. The question here is what metrics for perceptual video quality can improve the feedback from VLMs. Because the criteria of videos preferred by humans are multi-objective, maximizing a single metric may lead to undesirable generation due to over-optimization. For instance, focusing exclusively on temporal consistency or frame-by-frame quality metrics can unintentionally reduce the video's motion magnitude (see Appendix H). In this section, we first review the possible video quality metrics (Section 4.1), evaluate the Pearson correlation between these and the VLM feedback score, and then propose a reward calibration (Section 4.2), aiming to align the existing video rewards to VLMs by considering their weighted linear combination through the brute-force search of coefficients.

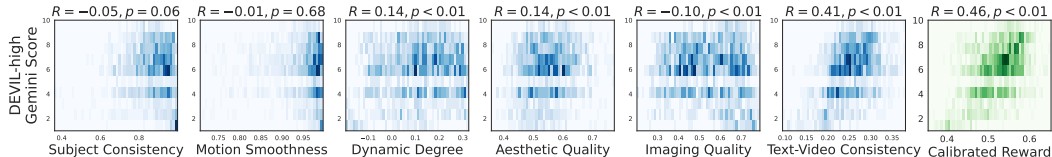

Figure 3: 2D-histogram and correlation between reward functions for perceptual video quality [12] and AI feedback from Gemini [14]. A single reward (e.g., subject consistency; blue) is often not aligned well with a preference from Gemini, which happens for all the prompt sets with different dynamics grades (see Figure 13). The calibrated reward, a linear combination of perceptual metrics via brute-force search (green), achieves the best Pearson correlation coefficient in all settings (statistically significant with $p < 0.01$).

## 4.1 Metric Reward for Perceptual Video Quality

Following Huang et al. [12], we select six base reward functions for perceptual video quality (see Appendix C):

- **Subject Consistency** quantifies how consistently the subject appears across video frames with DINO [47].
- **Motion Smoothness** leverages the motion prior in AMT [48] to evaluate whether the generated video's motion is smooth and physically plausible.
- **Dynamic Degree** quantifies the overall magnitude of dynamic object movement by estimating optical flow [49] for each pair of consecutive frames.
- **Aesthetic Quality** measures compositional rules, color harmony, and the overall artistic merit of each video frame with LAION aesthetic predictor [50].
- **Imaging Quality** assesses low-level distortions (e.g., over-exposure, noise, blur) in each frame with MUSIQ predictor [51].
- **Text-Video Consistency** captures how closely the content in a video aligns with a prompt with ViCLIP [52].

**Reward Calibration** To reflect the multi-dimensional aspect of preferred videos, we model the calibrated reward function $r^*(\cdot, \cdot)$ as a weighted linear combination of video quality metrics: $r^*(\mathbf{x}_0, \mathbf{c}) := \sum_{i=1}^{M} w_i r_i(\mathbf{x}_0, \mathbf{c}) / \sum_{i=1}^{M} w_i$. The coefficient $w_i$ is determined by maximizing the Pearson correlation with preference feedback. We heuristically conduct a brute-force search within a reasonable range (Section 4.2).

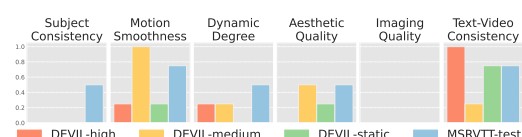

Figure 4: The coefficient of calibrated reward $w_i$ with feedback from Gemini. Each set of prompts, which has a different dynamics grade, requires a distinct mixture of perceptual video qualities.

**Experimental Setup** We leverage Gemini-1.5 [14] and GPT-4o [13] as automated raters for generated videos. We provide a prompt and generated video as inputs, instructing VLMs to assign discrete scores (from 1 to 10) based on overall visual quality (e.g., clarity, resolution, brightness, and aesthetic appeal), the appropriateness of motion for either static or dynamic scenes, the smoothness and consistency of shapes and motions, and the degree of alignment with a prompt (see Appendix I).

We select four prompt sets from two distinct datasets (see Appendix G). DEVIL [53] classifies its prompts into five categories depending on the dynamics grade, each further divided by subject type (e.g., cat, horse, truck, nature, etc.). We focus on three of the five dynamics grades (high, medium, and static) and select one prompt randomly from each subject-subdivision within a chosen category. We also draw 30 random captions from the test split of MSRVTT [54], widely used as a video benchmark.

We generate 64 videos per prompt from pre-trained Latte [18] using the DDIM sampler with $T = 50$ and $\eta = 0.0$ to examine the correlation among AI feedback and perceptual quality metrics. We also prepare candidates for the calibrated reward by choosing the combination of weights $w_i \in \{0.0, 0.25, 0.5, 0.75, 1.0\}$ and use those later to rank them based on the correlation with AI feedback.

## 4.2 Correlation and Reward Calibration

Figure 3 illustrates the 2D-histogram and the corresponding correlation between each metric and feedback from Gemini. Relying on a single metric often yields low correlation, which supports

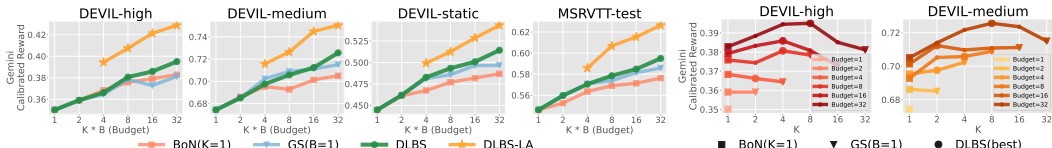

Figure 5: (**Left**) Comparison among diffusion latent beam search (DLBS), best-of-N (BoN), and greedy search (GS). We measure the performance in terms of a combinational reward calibrated to Gemini. DLBS improves all the calibrated rewards the best as the search budget $KB$ increases (especially $KB = 16, 32$), while BoN and GS, in some cases, eventually slow down or saturate the performance. Notably, an LA estimator with a small search budget ($KB = 8, T' = 6$) is comparable to or even outperforms DLBS ($KB = 32$). (**Right**) Optimal balance between the number of latent $K$ and the number of beams $B$ under the same budget. For instance, as we increase the budget to $KB = 16, 32$, we peak around $K = 4, 8, 16$, which is about 25–50% of the budget.

the multifaceted nature of perceptual video quality. See Appendix J.1 for further results, where we can see that the relative importance of each metric depends on the dynamics grade of the prompts; in highly dynamic DEVIL-high, the dynamic degree correlates more strongly with VLMs than consistency metrics. Conversely, subject consistency and motion smoothness play more prominent roles in less-dynamic DEVIL-medium or DEVIL-static. Because the aesthetic score focuses on frame-by-frame visual quality, it tends to correlate strongly with VLM in low-motion scenarios. In high-motion scenarios, in contrast, rapid movements and frequent transitions often introduce motion blur or abrupt changes in composition, reducing the frame-level aesthetic quality and thus weakening its correlation with VLMs.

Reward calibration, a weighted linear combination of these metrics, yields the highest correlation with Gemini (Figure 3, green). We select the best coefficients among brute-force candidates, based on the correlation with Gemini, for each set of prompts with a different dynamics grade (Figure 4). Prompts with a high dynamics grade, i.e., DEVIL-high, place greater weight on the dynamic degree. In contrast, prompts that describe slight motion, i.e., DEVIL-medium and DEVIL-static, place a smaller weight on it. In addition, Appendix L presents results from best-of-64 sampling with a single metric or calibrated reward, where a single metric often leads to over-optimization. This highlights the importance of reward calibration, appropriately weighting multiple criteria, as aligning with a request in the prompt.

## 5 Inference-Time Text-to-Video Alignment

**Experimental Setup** We use the same prompts and Gemini-/GPT-calibrated rewards as in Section 4. We compare the following inference-time search methods with a noise level $\eta = 1.0$ for DDIM:

- **Best-of-N Sampling (BoN):** We initialize $B$ latents and they follow the reverse process independently ($K = 1$). At $t = 0$, we evaluate the reward and select the best.
- **Greedy Search (GS):** At each denoising step, we select the best-rewarded diffusion latent ($B = 1$) from $K$ candidates sampled in a reverse process.
- **DLBS:** Given the budget $KB$, we sweep possible combinations in terms of power of 2 (e.g., $K = 8, B = 2$), and report the best results except for the case with $K = 1$ and $B = 1$.
- **DLBS-LA:** We combine DLBS with a lookahead estimator from the 6-step deterministic DDIM.

Our experiments aim to assess: (1) scaling the search budget and computational costs for efficient resource allocation (Section 5.1); (2) evaluating alignment performance with feedback from humans and VLMs (Section 5.2); (3) assessing scalability to capable SoTA models (Section 5.3); (4) quantitative analysis on the diversity of generated video (Section 5.4); (5) validating that DLBS is complementary to fine-tuning methods (Section 5.5); (6) performing detailed ablations on LA steps $T'$, and robustness to diverse and complex prompts (Section 5.6).

### 5.1 Scaling Search Budget and Computational Cost

Figure 5 (**Left**) measures the combinatorial reward calibrated to Gemini while increasing the search budget $KB \in \{1, 2, 4, 8, 16, 32\}$. DLBS improves all the calibrated rewards the best as $KB$ increases (especially $KB = 16, 32$), while BoN and GS eventually slow down or saturate the performance in some cases. See Appendix M.1 for results with GPT calibrated reward and Appendix M.2 for the results with $KB = 64$, where we still observe the improvement.

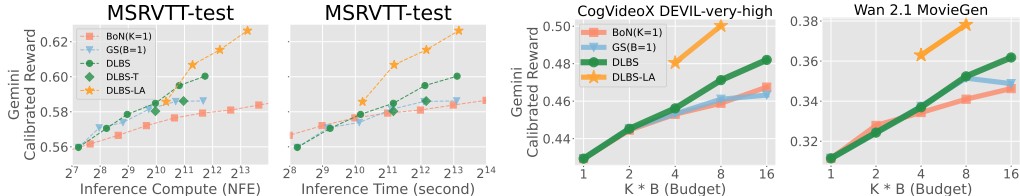

Figure 6: (**Left**) DLBS achieves alignment performance gains more efficiently than BoN and GS under the same number of function evaluations (NFE) or execution time. Increasing the search budget provides larger improvements under an equivalent computational cost than scaling the number of diffusion steps (DLBS-T; $KB = 8, T = 100, 200$). Employing the LA estimator (DLBS-LA) further amplifies these gains, only with marginal overhead, yielding remarkably better efficiency than BoN or GS. (**Right**) DLBS and DLBS-LA help the latest SoTA models, CogVideoX-5B [19] and Wan 2.1-14B [20], improve the generated video quality.

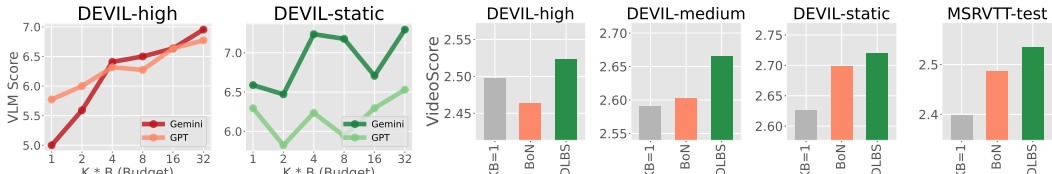

Figure 7: (**Left**) DLBS on calibrated reward can improve the original preference feedback from VLMs. As in Figure 5, we use each calibrated reward (Figure 4) for DLBS and evaluate the quality with Gemini or GPT-4o. (**Right**) DLBS on calibrated reward also improves another qualitative metric, the most, VideoScore [55], which is not involved in a reward calibration.

Figure 5 (**Right**) demonstrates the scaling trend of DLBS, proportional to the search budget, under various choices of $K$. The results show that there is an optimal balance between the number of latent $K$ and the number of beams $B$ under the same budget. For instance, as we increase the budget to $KB = 16, 32$, we peak around $K = 4, 8, 16$, which is about 25–50% of the budget. This implies that balancing possession and exploration of diffusion latents in DLBS helps search for the best outputs robustly. See Appendix M.3 for further results.

We also analyze how alignment performance scales with different DLBS configurations under fixed computational budgets, using the number of function evaluations (NFE) and wall-clock time as cost measures. As shown in Figure 6 (**Left**), DLBS consistently outperforms BoN and GS across both budgets, demonstrating superior efficiency in utilizing compute. Adding the LA estimator further amplifies this advantage, offering substantial performance gains with minimal overhead. In contrast, increasing the number of diffusion steps $T$ (DLBS-T; $KB = 8, T \in \{100, 200\}$) results in only marginal improvements despite the higher computational cost. Our results suggest a clear strategy for inference-time budget allocation: prioritize enabling the LA estimator and increasing the search budget $KB$ leads to substantial performance gains, while increasing the diffusion steps $T$ provides limited benefit relative to its computational cost.

## 5.2 Evaluation with AI and Human Feedback

As discussed in Section 4, we obtain a manageable reward function through the reward calibration, which reduces the cost for frequent evaluation queries in inference-time search. While DLBS efficiently improves the calibrated reward (Figure 5; **Left**), a natural question is whether DLBS can improve an actual assessment by VLMs or humans by optimizing their calibrated rewards. We first use each calibrated reward for DLBS, then evaluate the quality using discrete scores (from 1 to 10) from Gemini or GPT-4o. Figure 7 (**Left**) demonstrates that DLBS maximizing calibrated rewards can improve the original preference feedback from VLMs, as we grow the search budget.

Next, we evaluate with VideoScore [55], a metric trained on human judgments that evaluates videos across five quality criteria. Figure 7 (**Right**) shows that DLBS significantly improves the quantitative evaluation based on human evaluation. Lastly, we perform pairwise comparisons between DLBS-LA ($KB = 8, T' = 6$, NFE = 2500) and BoN ($KB = 64$, NFE = 3200) by three human evaluators (Figure 8; **Left**). The results confirm that, whatever models or prompts we choose, the quality of content generated by DLBS-LA consistently outperforms that of a baseline despite requiring fewer NFEs. This emphasizes that our proposed method, integrating reward calibration and beam search in a latent space, effectively enhances perceptual video quality. See Appendix M.11 for the details.

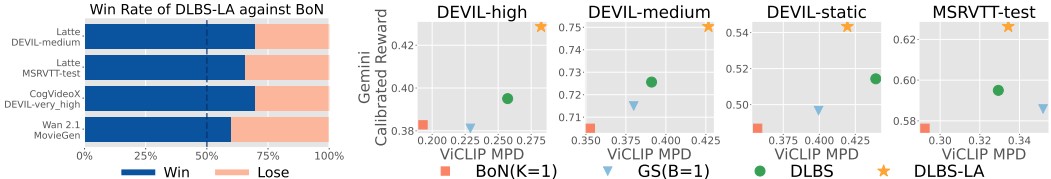

Figure 8: (**Left**) The pairwise comparisons of DLBS-LA ($KB = 8$, $T' = 6$, NFE = 2500) and BoN ($KB = 64$, NFE = 3200) by three human raters, when DLBS-LA searches effectively for the best latents in a diffusion process. (**Right**) Alignment–diversity tradeoff ($KB = 32$). The mean pairwise distance (MPD) of ViCLIP embeddings is used as a measure of diversity. DLBS and DLBS-LA ($T' = 6$) achieve high performance while maintaining higher diversity.

Table 1: Performance of DLBS with DPO finetuned VideoCrafter2 on DEVIL-high and MSRVTT-test datasets. While DPO alone yields marginal improvements, combining it with DLBS leads to notable gains, demonstrating the compatibility of inference-time search with fine-tuning approaches.

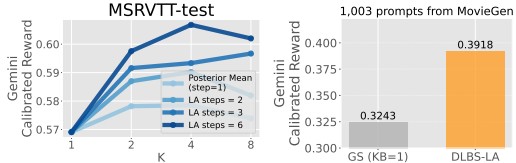

Figure 9: (**Left**) Comparison of different LA steps $T'$ on MSRVTT-test ($KB = 8$). The performance improves as the number of LA steps increases. (**Right**) Perceptual quality comparison on a large and complex prompt set (1,003 prompts from Movie Gen Video Bench [57]). DLBS-LA can generalize to diverse prompts.

| Method | DEVIL-high | MSRVTT-test |
|---|---|---|
| VideoCrafter2 | 0.337 | 0.555 |
| + DPO | 0.335 | 0.556 |
| + DPO & DLBS | **0.359** | **0.576** |

## 5.3 Scaling Model Parameters and Capabilities

We scale up the base diffusion model from Latte to the latest SoTA models, such as CogVideoX-5B [19] and Wan 2.1-14B [20], and evaluate if DLBS can improve the generation quality from those larger models. Note that we use SDE-DPMSolver++ [56] for Wan 2.1 experiments. Since base models are more capable here, we adopt more challenging prompts from DEVIL-very-high (22 prompts) and Movie Gen Video Bench [57] (20 prompts). See Appendix G and Appendix J.2 for the details. Figure 6 (**Right**) shows that our methods achieve significant improvements in calibrated reward for both models. As shown in Figure 8 (**Left**), human evaluation also supports that our proposed method could generally work well with any text-to-video models, even with more capable models in the future. See Appendix M.9 for results with a maximum frame length, and Appendix M.10 for further results with CogVideoX and Wan 2.1.

## 5.4 Alignment-Diversity Tradeoff

Alignment for diffusion models can steer desirable outputs, but it is said that the diversity of generated samples or the performance of original models often degrade [10, 37]. While inference-time search does not change or degrade the model itself, we here compare the diversity of samples among BoN, GS, DLBS, and DLBS-LA. We measure the sample diversity as the mean pairwise distance of ViCLIP [52] embeddings (see Appendix D). Figure 8 (**Right**) reveals that DLBS and DLBS-LA achieve high performance with higher diversity than BoN or GS. This exhibits a benefit from the wider possession and exploration of diffusion latents in DLBS and DLBS-LA.

## 5.5 DLBS is Compatible with Finetuning

In image generation, Ma et al. [17] has shown that allocating additional computation at inference time can be more effective than relying solely on post-training approaches. We find a similar trend in video generation. To examine this, we apply a representative fine-tuning method, VideoDPO [58], to VideoCrafter2 [3]. As shown in Table 1, VideoDPO alone brings negligible improvement over the baseline. However, when combined with DLBS, the performance increases substantially on both DEVIL-high and MSRVTT-test. These results indicate that DLBS is complementary to post-training methods, enabling further performance gains even after fine-tuning. See Appendix M.7 and Appendix M.8 for results comparing DLBS with other alignment methods.

## 5.6 Ablation Study

**Lookahead Steps for Reward Estimate** We scale up LA steps $T'$ to obtain an accurate reward estimate. We use MSRVTT-test ($KB = 8$) and Gemini reward for experiments. Figure 9 (**Left**) shows that as the number of LA steps increases, the performance improves more. Even $T' = 2, 3$ significantly outperforms the posterior mean, which is often used in prior works [15, 11, 41]. This is because the sub-optimal performance of inference-time search comes from the approximation errors, and LA estimator can notably reduce them. As shown in Figure 5 (**Left**), DLBS-LA ($KB = 8, T' = 6$) achieves comparable or even outperforming results with DLBS ($KB = 32$). It is quite beneficial to spend a computation to estimate the reward accurately. See Appendix M.4 for further discussions. In addition, an ablation study on diffusion steps is shown in Appendix M.5.

**A Large and Complex Prompt Set** To confirm the robustness of our approach on a large and highly diverse prompt set, we compare GS ($KB=1$) with DLBS-LA ($KB=8$, $T'=6$) with all the prompts in Movie Gen Video Bench [57], which comprises 1,003 prompts (Figure 9; **Right**). We observe that DLBS-LA consistently delivered substantially higher alignment rewards, demonstrating that DLBS generalizes effectively to complex prompt distributions. Additionally, we also assess reward transferability by applying weights calibrated on DEVIL-high and DEVIL-medium to MSRVTT-test prompts, consistently improving scores (see Appendix J.4).

## 6 Related Works

Classifier guidance [35, 59] has been the most popular to enhance text-content alignment. On top of that, recent works [11, 39] leverage reward or external feedback at inference time by selecting better latents [60], which probably achieve higher rewards during the reverse process. Kim et al. [61] propose twisted SMC [37] with reward gradient, which is not suitable for non-differentiable feedback and for domains such as video, where reward gradient needs a huge memory cost. Gradient-free methods [62, 17] such as SMC [41] or greedy search [15] often exhibit sub-optimal results affected by inaccurate reward estimates from noisy latents. Yeh et al. [63] uses ODE to estimate the reward, but it highly depends on Karras sampler [28] to avoid numerical instability. In contrast, we address the error propagation from inaccurate reward estimates with beam search and lookahead estimator via deterministic DDIM, which is more popular and stable. Our methods work more scalably when allocating more computation budget at inference time. See Appendix N for further related works.

## 7 Discussion and Limitation

Our reward calibration assumes that VLMs serve as a proxy for human evaluation, and we demonstrate both qualitative and quantitative improvements in video quality through evaluations by VLMs and human raters. In future work, incorporating more specialized and accurate evaluators (e.g., reward models that focus on physical laws [9]) could enable a more fine-grained analysis. In practice, we often do implicit or explicit best-of-N sampling for video generation. In contrast, DLBS-LA exhibits much better computational efficiency. Spending more computation at inference time significantly improves perceptual quality, but it is orthogonal compared to speeding up the sampling process via distillation [64, 65], architecture changes [66], or parallel sampling [67]. We believe both high-quality and speedy sampling have practical needs and should be balanced.

## 8 Conclusion

This paper studies which metrics we should optimize and how to optimize them for better text-to-video generation. We point out that feedback from humans or capable VLMs reflects multiple dimensions of video quality, so optimizing an existing metric alone is insufficient; rather, we should calibrate the reward by combining. Our DLBS with LA estimator reduces the error propagation from the inaccurate reward estimate. We demonstrate that DLBS is the most scalable, efficient, and robust inference-time search that significantly improves video quality under the same computational costs. We hope our work encourages more uses of inference-time computation for text-to-video models.

## Acknowledgements

We thank Po-Hung Yeh, Shohei Taniguchi, Kuang-Huei Lee, Arnaud Doucet, Heiga Zen, Robin Scheibler, and Yusuke Iwasawa for their support and helpful discussion on the initial idea of this work. HF was supported by JSPS KAKENHI Grant Number JP22J21582 (by March 2025), and MS was supported by JSPS KAKENHI Grant Number JP23H04974. We also appreciate the funding support from Google Japan.

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

# A Broader Impacts

Our work contributes to the progress of text-to-video by focusing on improving the perceptual quality and fidelity of generated videos, specifically addressing issues like unnatural movement, deformation, and temporal inconsistencies, through the inference-time alignment algorithm. Such advancements hold immense potential for revolutionizing creative fields and enabling new applications in gaming, filmmaking, and robotics.

On the other hand, the ability to generate highly realistic videos raises concerns about the potential for misuse in creating deceptive content, including deepfakes and misinformation. Like other generative models, such as large language models, text-to-video models, and their inference-time search, may inherit and amplify biases present in the training data due to the misalignment. This might lead to the generation of videos that perpetuate harmful stereotypes or underrepresent certain groups.

Lastly, by focusing on inference-time alignment, our method promotes more use of computational resources at test time. On one side, this may increase the environmental footprint for running large generative models and on the other side, our detailed recipe can contribute to designing efficient use of resources and reducing the footprint associated with training. We believe that discussing this aspect is crucial as the scale of these models continues to grow.

# B Implementation Details

**Code**  Our implementation for the experiments are available at `https://github.com/shim0114/T2V-Diffusion-Search`.

**Models**  Our experiments cover three text-to-video diffusion models.

- **Latte** [18]: a T5-conditioned latent diffusion transformer with 1.1 B parameters, built on PixArt-$\alpha$ [68, 69].
- **CogVideoX** [19]: a larger DiT-based model with 2B or 5B parameters. We mainly used CogVideoX-5B.
- **Wan 2.1** [20]: a DiT-based flow model with 1.3B or 14B parameters. We use Wan 2.1-1.3B for reward calibration and Wan 2.1-14B for inference-time search experiments.

**Hyperparameters**

- **Latte**: DDIM scheduler with a linear noise schedule ($\beta_{\text{start}} = 1.0 \times 10^{-4}$, $\beta_{\text{end}} = 2.0 \times 10^{-2}$) and classifier-free guidance scale $w_{\text{cfg}} = 7.5$.
- **CogVideoX**: DDIM scheduler with the original settings and $w_{\text{cfg}} = 6.0$. Owing to computational resource constraints, we limit the frame length to 17 per sample.[3]
- **Wan 2.1**: DPMSolver++ with guidance scale $w_{\text{cfg}} = 5.0$. For the same computational reasons, the spatial resolution is limited to $832 \times 480$ and the frame length to 33 per sample.[3]

**Hardware configuration**

- **Latte**: FP16 inference on a single NVIDIA A100 (40 GB), batch size 1.
- **CogVideoX**: BF16 inference on a single NVIDIA A100 (40 GB), batch size 1.
- **Wan 2.1**: FP16 inference on four NVIDIA H100s (80 GB each), batch size 1.

**AI-feedback endpoints**  We use API endpoints: `gemini-1.5-pro-002` and `gpt-4o-2024-11-20`.

---

[3]The experiments reported in Appendix M.9 were conducted with a different number of frames.

## C  Details of Metric Rewards

**Subject Consistency**  We adopt the subject consistency metric proposed in VBench [12] to quantify how consistently a subject is depicted across consecutive video frames. Concretely, for each frame $i$ in a video, we extract a feature representation $\mathbf{d}_i$ using DINO [47] with a ViT-B/16 [70] backbone. Let $\langle \mathbf{d}_i, \mathbf{d}_j \rangle$ denote the cosine similarity between the features $\mathbf{d}_i$ and $\mathbf{d}_j$. Then, VBench defines the subject consistency metric as follows:

$$R_{\text{subject}} = \frac{1}{T-1} \sum_{t=2}^{T} \frac{1}{2} \big( \langle \mathbf{d}_1, \mathbf{d}_t \rangle + \langle \mathbf{d}_{t-1}, \mathbf{d}_t \rangle \big). \tag{8}$$

DINO, which is trained in a self-supervised manner using unlabeled images and image augmentations, does not explicitly suppress intra-class variations. As a result, it remains particularly sensitive to identity shifts within the same subject, making it well-suited for evaluating subject consistency across frames.

**Motion Smoothness**  We adopt the frame-interpolation-based metric originally proposed in VBench [12] to assess whether a generated video's motion is smooth and physically plausible. In particular, this metric leverages the motion prior from AMT [48], employing its AMT-S variant for frame reconstruction. Concretely, let $[\mathbf{f}_0, \mathbf{f}_1, \mathbf{f}_2, \ldots, \mathbf{f}_{2n}]$ denote the frames of a generated video. We remove each odd-numbered frame to obtain a lower-frame-rate sequence $[\mathbf{f}_0, \mathbf{f}_2, \mathbf{f}_4, \ldots, \mathbf{f}_{2n}]$, and rely on AMT-S to reconstruct the missing frames $[\hat{\mathbf{f}}_1, \hat{\mathbf{f}}_3, \ldots, \hat{\mathbf{f}}_{2n-1}]$. We then compute the Mean Absolute Error (MAE) between these reconstructed frames and the original odd-numbered frames, denoting this measure by $R_{\text{smoothness}}$. Finally, following the normalization scheme introduced in VBench, we define:

$$R_{\text{smoothness-norm}} = \frac{255 - R_{\text{smoothness}}}{255}, \tag{9}$$

which ensures that the final score lies in the range $[0, 1]$, with higher values indicating smoother motion. This measure leverages the motion prior in AMT to evaluate whether the generated video's motion is smooth and physically plausible. We remove each odd-numbered frame, then use AMT-S to reconstruct those frames based on short-term motion assumptions.

**Dynamic Degree**  This measure quantifies the overall magnitude of dynamic object movement. Let $T$ be the total number of frames in the generated video. For each pair of consecutive frames $t$ and $t+1$, we estimate the optical flow $\mathbf{v}_t$ using RAFT [49], compute its norm $\|\mathbf{v}_t\|$, and sum these values across all frames:

$$R_{\text{dynamics}} = \sum_{t=1}^{T-1} \|\mathbf{v}_t\|. \tag{10}$$

We then apply a logarithmic transformation to $R_{\text{dynamics}}$ and divide by 16:

$$R_{\text{dynamics-rescaled}} = \frac{\log\big(R_{\text{dynamics}}\big)}{16}. \tag{11}$$

This rescaling helps ensure that the value range of $R_{\text{dynamics-scaled}}$ is roughly comparable to other metrics in our evaluation.

**Aesthetic Quality**  This criterion evaluates compositional rules, color harmony, and overall artistic merit on a per-frame basis. Concretely, for each frame $i$ in a video, we extract a CLIP image embedding $\mathbf{c}_i^{image}$ using the CLIP ViT-L/14 model [71]. We then feed $\mathbf{c}_i^{image}$ into the LAION aesthetic predictor [50], which assigns a raw rating $r_i \in [0, 10]$. To normalize these scores to the $[0, 1]$ range, we set

$$r_i' = \frac{r_i}{10}. \tag{12}$$

Let $T$ be the total number of frames. The final aesthetic reward is then obtained by taking the average of the normalized ratings across all frames:

$$R_{\text{aesthetic}} = \frac{1}{T} \sum_{i=1}^{T} r_i'. \tag{13}$$

Because the LAION aesthetic predictor leverages CLIP embeddings instead of raw images, it captures higher-level features related to composition, color harmony, and artistic appeal.

**Imaging Quality** This indicator assesses low-level distortions (e.g., over-exposure, noise, blur) in each generated frame. We adopt the MUSIQ predictor [51], trained on the SPAQ dataset [72]. The frame-wise score is normalized to [0, 1] by dividing by 100, and the final video score is the mean of these normalized values across all frames in the same way as Equation 12 and Equation 13.

**Text-Video Consistency** This measure captures how closely a generated video's content aligns with its text prompt. We employ ViCLIP [52], a model pre-trained on a 10M video-text dataset and fine-tuned to handle temporal relationships, to embed both the video frames and the text. Since ViCLIP computes embeddings from 8-frame inputs, we sample 8 frames from each video. Let $\mathbf{v}^{\text{video}}$ denote the resulting video embedding and $\mathbf{v}^{\text{text}}$ denote the text embedding. We then define the final alignment score as the cosine similarity between these embeddings:

$$R_{\text{tv-consistency}} = \langle \mathbf{v}^{\text{video}}, \mathbf{v}^{\text{text}} \rangle \tag{14}$$

## D   Details of Sample Diversity

We measure the sample diversity as the mean pairwise distance of ViCLIP [52] embeddings to quantify the diversity in videos, inspired by the approach for evaluating diversity in images [61]. Specifically, given $N$ generated video samples, we first extract ViCLIP embeddings $\mathbf{v}^{\text{video},(i)}$ for each sample $i$. The pairwise diversity score is then computed as the mean pairwise distance:

$$D_{\text{video-diversity}} = \frac{1}{N(N-1)} \sum_{i \neq j} \left( 1 - \langle \mathbf{v}^{\text{video},(i)}, \mathbf{v}^{\text{video},(j)} \rangle \right). \tag{15}$$

Here, $\langle \mathbf{v}^{\text{video},(i)}, \mathbf{v}^{\text{video},(j)} \rangle$ denotes the cosine similarity between the ViCLIP embeddings of two generated videos $i$ and $j$. This formulation is similar to Equation 14, but in the case of pairwise distance computation, we take the pairwise mean of $1 - (\text{cosine similarity})$ to obtain a diversity measure.

# E   Algorithms with Continuous-time Diffusion Process

In this section, we present our algorithms for a continuous-time diffusion process. For Wan 2.1 [20], we integrated the proposed search algorithm into DPMSolver++ [56], a widely used continuous-time solver. Algorithms 3 and 4 present the pseudocode. Although we present the first-order variant for clarity, the procedure extends straightforwardly to higher-order formulations. Throughout this section, we adopt the notation of Lu et al. [56].

---

**Algorithm 3** Diffusion Latent Beam Search (DLBS) with SDE-DPMSolver++

---

**Input:** signal prediction latent diffusion model $z_\theta$, reward function $r'$, time steps $\{t_s\}_{s=0}^{M}$, noise scheduling parameter $\{\alpha_{t_s}\}_{s=0}^{M}, \{\sigma_{t_s}\}_{s=0}^{M}$, number of beams $B$, number of candidates $K$

1: $\mathbf{z}_{t_0}^1, \cdots, \mathbf{z}_{t_0}^B \sim \mathcal{N}(\mathbf{0}, \mathbf{I})$  ▷ Initial $B$ beams
2: **for** $s = 1$ **to** $M$ **do**
3:  **for** $j = 1$ **to** $B$ **do**
4:   ▷ Update one step to produce $\mathbf{z}_{t_s}^j$
5:   $\mathbf{z}_{t_s}^j = \frac{\sigma_{t_s}}{\sigma_{t_{s-1}}} e^{-h} \mathbf{z}_{t_{s-1}}^j + \alpha_{t_s}(1 - e^{-2h}) z_\theta(\mathbf{z}_{t_{s-1}}^j)$
6:  **end for**
7:  **if** $s < M$ **then**
8:   **for** $j = 1$ **to** $B$ **do**
9:    ▷ Sample $K$ next candidate latents
10:   $\mathbf{z}_{t_s}^{ij} = \mathbf{z}_{t_s}^j + \sigma_t \sqrt{e^{-2h} - 1} \epsilon_{t_{s-1}}^i$ with $\epsilon_{t_{s-1}}^1, ..., \epsilon_{t_{s-1}}^K \sim \mathcal{N}(\mathbf{0}, \mathbf{I})$
11:   ▷ Estimate the clean sample from noisy latent
12:   $\hat{\mathbf{z}}_{t_M|t_s}^{ij} = \frac{\sigma_{t_M}}{\sigma_{t_s}} \mathbf{z}_{t_s}^{ij} - \alpha_{t_M}(e^{-h} - 1) z_\theta(\mathbf{z}_{t_s}^{ij}))$
13:  **end for**
14:  ▷ Search $B$ higher-reward beams from $KB$ latents
15:  $\texttt{budget} := \{(\mathbf{z}_{t_s}^{11}, \hat{\mathbf{z}}_{t_M|t_s}^{11}), \cdots, (\mathbf{z}_{t_s}^{KB}, \hat{\mathbf{z}}_{t_M|t_s}^{KB})\}$
16:  **for** $j' = 1$ **to** $B$ **do**
17:   $\mathbf{z}_{t_s}^{j'} = \text{argmax}_{\mathbf{z}_{t_s}^{ij} \in \texttt{budget}} \; r'(\hat{\mathbf{z}}_{t_M|t_s}^{ij})$
18:   $\texttt{budget} = \texttt{budget} \setminus \{(\mathbf{z}_{t_s}^{j'}, \hat{\mathbf{z}}_{t_M|t_s}^{\text{argmax}})\}$
19:  **end for**
20:  $j \in \{1, \cdots, B\} \leftarrow j'$  ▷ Reset selected $B$ indices
21:  **end if**
22: **end for**
23: **return:** $\mathbf{z}_{t_M} = \text{argmax}_{\mathbf{z}_{t_M}^j \in \{\mathbf{z}_{t_M}^1, \cdots, \mathbf{z}_{t_M}^B\}} \; r'(\mathbf{z}_{t_M}^j)$

---

**Algorithm 4** Lookahead (LA) with DPMSolver++

---

**Input:** signal prediction latent diffusion model $z_\theta$, current diffusion latent $\mathbf{z}_{t_s}$, number of lookahead steps $M'(<< M)$

1: ▷ Run $M'$-step deterministic DPMSolver++ starting from $\mathbf{z}_{t_s}$
2: $\tilde{s}(u) \in \{s, \ldots, \lfloor \frac{M'-u}{M'}s + \frac{u}{M'}M \rfloor, \ldots, \lfloor \frac{1}{M'}s + \frac{M'-1}{M'}M \rfloor, M\}$
3: Select new lookahead noise schedule $\{\tilde{\alpha}_{t_u}\}_{u=0}^{M'}$ for $M'$-**step interpolation** of the rest of original $\{\alpha_{t_{s'}}\}_{s'=0}^{M}$
4: $\mathbf{z}_{t_{\tilde{s}(0)}} := \mathbf{z}_{t_s}$
5: **for** $u = 1$ **to** $M'$ **do**
6:  $\tilde{\mathbf{z}}_{t_{\tilde{s}(u)}|t_{\tilde{s}(u-1)}} = \frac{\sigma_{t_{\tilde{s}(u)}}}{\sigma_{t_{\tilde{s}(u-1)}}} \mathbf{z}_{t_{\tilde{s}(u-1)}}^{ij} - \alpha_{t_{\tilde{s}(u)}}(e^{-h} - 1) z_\theta(\mathbf{z}_{t_{\tilde{s}(u-1)}}^{ij}))$
7: **end for**
8: **return:** $(\mathbf{z}_{t_s}, \tilde{\mathbf{z}}_{t_M|t_{\tilde{s}(0)}})$ ▷ Latent and LA estimator

---

# F Theoretical Analysis on Lookahead Estimator

Consider the Lookahead estimator described in Algorithm 2, which obtains the state $\tilde{\mathbf{z}}_{0|\tilde{t}(0)}$ by performing $T'$ steps of DDIM (or another diffusion-based sampling) with $\eta = 0.0$. Our goal is to show that, as $T'$ grows, the reward estimate $r'(\tilde{\mathbf{z}}_{0|\tilde{t}(0)})$ converges to $r'(\mathbf{z}_0)$, thereby improving estimation accuracy.

Let $\mathbf{z}_{t-1}$ be a state in the latent space from which we wish to recover the initial latent $\mathbf{z}_0$. By applying $T'$ steps of DDIM with $\eta = 0.0$, we obtain an approximation $\tilde{\mathbf{z}}_{0|\tilde{t}(0)}$. From prior work [73], the error $\|\tilde{\mathbf{z}}_{0|\tilde{t}(0)} - \mathbf{z}_0\|$ scales as follows:

$$\|\tilde{\mathbf{z}}_{0|\tilde{t}(0)} - \mathbf{z}_0\| \leq \begin{cases} \mathcal{O}(1/T') & \text{(DDIM)}, \\ \mathcal{O}(1/\sqrt{T'}) & \text{(DDPM)}, \\ \mathcal{O}(1/(T')^n) & \text{(an } n\text{-th order solver)}. \end{cases}$$

Hence, increasing $T'$ yields a progressively better approximation of $\mathbf{z}_0$.

Assume $\mathbf{z}_0$ is the latent representation at time $t = 0$. By the Continuous Mapping Theorem, if $\tilde{\mathbf{z}}_{0|\tilde{t}(0)} \to \mathbf{z}_0$ as $T' \to \infty$, then for any continuous function $f$, we have

$$f(\tilde{\mathbf{z}}_{0|\tilde{t}(0)}) \;\to\; f(\mathbf{z}_0).$$

Setting $f(\cdot) = r'(\cdot)$, where $r'$ is our reward model, yields

$$r'(\tilde{\mathbf{z}}_{0|\tilde{t}(0)}) \;\to\; r'(\mathbf{z}_0),$$

as $T' \to \infty$.

We further assume that the reward model $r'(\cdot)$ is Lipschitz continuous with Lipschitz constant $L$. Then for any two latent states $\mathbf{z}_a$ and $\mathbf{z}_b$, the reward estimates satisfy

$$|r'(\mathbf{z}_a) - r'(\mathbf{z}_b)| \;\leq\; L\|\mathbf{z}_a - \mathbf{z}_b\|.$$

Hence, the order of the error in $r'(\tilde{\mathbf{z}}_{0|\tilde{t}(0)})$ tracks the order of the error in $\tilde{\mathbf{z}}_{0|\tilde{t}(0)}$ itself. Explicitly,

$$\left| r'(\tilde{\mathbf{z}}_{0|\tilde{t}(0)}) - r'(\mathbf{z}_0) \right| \;\leq\; L\,\|\tilde{\mathbf{z}}_{0|\tilde{t}(0)} - \mathbf{z}_0\|,$$

implying that an $\mathcal{O}(1/T')$ (or better) approximation in latent space implies an $\mathcal{O}(1/T')$ (or correspondingly better) approximation in the reward space.

As $T'$ increases, $\tilde{\mathbf{z}}_{0|\tilde{t}(0)}$ converges to $\mathbf{z}_0$, and consequently $r'(\tilde{\mathbf{z}}_{0|\tilde{t}(0)})$ converges to $r'(\mathbf{z}_0$. Because the reward model is Lipschitz continuous, this convergence ensures that the error in reward estimation decreases at the same order as the error of the latent approximation. Therefore, employing the LA estimator with a larger $T'$ yields a more accurate reward estimate.

# G    List of Prompts

**MSRVTT-test**

1. a woman is singing on stage about that one person being the one she wants
2. someone is filming a parked car in the parking lot
3. a cat is feed it s babies and a rabbit
4. mario game with bombs
5. someone is browsing a set of games on their console
6. a game is being played
7. a man holds a very large stick
8. a yellow-haired girl is explaining about a game
9. a ship is sailing around on the water
10. a woman with blonde hair and a black shirt is talking
11. a buffalo is attacking a man
12. a band is playing music and people are dancing
13. a child is playing a video game
14. a person is showing how to fold paper
15. a woman is sitting down on a couch in a room
16. a man inside of a car is using his finger to point
17. a man waters his plants
18. the symmetrical cone is japan s most famous symbol
19. an indoor soccer game
20. a japanese monkey bathing in a hot spring with pleasant music
21. some images of motorcycles are being shown on tv
22. someone is serving food in the restaurand
23. this is a competition type show
24. a woman on the news is talking about a story
25. this is a phone review video
26. some fake horses are standing around in a game
27. a person is filming a white car interior seat
28. video of clips from a movie
29. a man with a blue and white shirt is walking around
30. person making something in the kitchen

**DEVIL-high**

1. A bookshelf collapses loudly, books flying everywhere, creating chaos in the once quiet room.
2. Swift scenes of a sandstorm engulfing a desert oasis, with dunes shifting and palm trees bending in the relentless wind.
3. A chaotic scene of cowboys rounding up cattle during a stampede.
4. Suddenly, a storm hits the city, rain pouring down like a torrent, making rivers on the streets.
5. WWI biplanes in a dogfight with canvas wings ripping, dramatic cloud backdrop, ultra-detailed.
6. In the mountains, a bear erupts from the snow, creating a large cloud of powder.
7. Amidst a thunderstorm, a lightning bolt strikes a bicycle, setting it ablaze with crackling energy and lighting up the dark, rainy street.
8. A single eagle dives extremely fast, snatching a fish from the water.
9. A boat hits a big wave and flips, landing upside down.
10. A car drives through a wall of fire in a daring escape.
11. The cat tore across the living room, jumping over toys and furniture to catch the mouse.
12. A cow jumps over a fence, landing in a pond with a big splash.
13. Two dogs chase each other, suddenly skidding around a sharp corner.
14. A storm sweeps an elephant into a raging river, carrying it away swiftly.
15. Racing the sunset, a giraffe charges across the horizon, shadows stretching long.
16. Against the wind, a lone horse gallops, mane streaming behind.
17. Jumping over a gorge, the motorcycle lands just in time on the other side.
18. A thief sprints away from the scene, with the police in hot pursuit.
19. The ice cracks beneath their feet, making the sheep skid and slide, rushing to solid ground.
20. Lightning strikes as a train blasts its horn, cutting through a stormy night.
21. A truck speeds across the desert, dust clouds swirling behind it.
22. Under a rainbow, a zebra kicks up a spray of water as it crosses a fast-flowing river.

## DEVIL-medium

1. London heathrow, united kingdom - 05 12 2019: 4k super-telephoto plane accelerates down hot runway through heat shimmer
2. A cool dj teddy bear with sunglasses on top of turntable with video static
3. Aerial view. cute girl in the coat drive on country road on the bicycle
4. Brown pelican flying flight in fall bay harbor in ecuador
5. Small fishing boat, anchored on a silver ocean, in thailand.
6. a filled yellow school bus with over-sized black wheels drives through a flooded area with red lights on and gets splattered with mud
7. St. petersburg, russia - circa march, 2015: vehicles drive on city ringroad at evening time. st. petersburg ring road is a main route encircling the city
8. cat manages to hang on to dangling object
9. Taking cow milk cheese with fork 4k footage
10. dog passes in and out of view
11. 1930s: elephant roars, man shoots at elephant. elephants walk through jungle. man tries to fire gun, throws gun on ground, runs away.
12. the baby giraffe is zoomed in on and then camera shakes
13. Cowboys drive group of horses at farming enterprise.
14. 4k couple watching film or tv at home & jumping with shock at the action
15. contestants are reading themselves to start a mini-motorbike race
16. Macao beach with stone mountains aerial view from drone. travel destination. summer vacation. dominican republic
17. Male boxer resting and sweating after boxing training
18. Wild tulips in a meadow on background sky. sunrise. bonfire. a quiet spring morning in the steppe.
19. Sheep eating grass in punata and potosi, bolivia.
20. Bodo arctic town norway - ca july 2018: train station building and rails tilt up
21. a woman is describing different sets of tubes and hoses in the back of a white pick up truck which is parked on the side of a street with cars going by in the background
22. Istanbul, turkey - october 2018: commuters inside istanbul metro wagon travelling towards taksim station

## DEVIL-static

1. airplane with red body is shown for first time.
2. a man holds up a stuffed bear.
3. when you can see the first view of the full bike
4. second bird lands on feedersecond bird lands on feeder
5. a red boat is first seen.
6. Tourist bus station 3d realistic footage. public transport front view animation. vehicles on modern urban highway bridge background. passengers transportation parking. city bus stop video
7. black car is under the blue sign.
8. cat looks at the camera
9. dog puts paws together
10. a white horse standing beside red colored wearing girl dress standing with stick bending down knee displaying on screen
11. Blurred conference room with audience - 4k video
12. first time we see orange branch to the right
13. A woman and a man. holding a gift.
14. A tranquil tableau of the old red barn stood weathered and iconic against the backdrop of the countryside
15. black numbers 1758 at bottomof train
16. a large white box truck travels through water is followed by two other trucks and ascends a gray road through mountains
17. view of big city from balconyview of big city from balcony

## DEVIL-very-high

1. Classical style of a horse partaking in an ancient chariot race, scenes switching quickly from cheering crowds to close-ups of intense wheel clashes.
2. High-speed shots of a volcanic eruption engulfing a tropical island, with lava fountains spewing molten rock and the environment transforming from idyllic paradise to hellish landscape of ash and fire.
3. A thrilling scene of rural mountain biking extreme sports, starting from the early morning cycling adventure, transitioning to the intense chase through fields and forests, and ending with cheers and celebrations under the sunset.
4. Neon-lit streets pulse with energy as vehicles engage in a high-octane pursuit, transitioning seamlessly from chaos to calculated evasion. Against the backdrop of a setting sun, the chase intensifies, each turn a heartbeat away from capture.
5. A fighter jet dodging rapid anti-air gunfire, quick maneuvers, tracer rounds visible.
6. Bear escaping a collapsing cave, rocks tumbling, dust rising, ((masterpiece)), ((best quality)), 8k, high detailed, ultra-detailed, bear, ((dark rocky textures)), sprinting, (echoing rumble), sudden movement
7. A courier on a bike weaves through traffic at breakneck speed, narrowly avoiding cars and pedestrians in a rush to make deliveries on time.
8. A hummingbird rapidly darting between vibrant flowers in a lush garden, with quick cuts to various close-up shots showcasing its rapid wing movement and agility.
9. Venetian gondola chase scene, narrow canals, historic buildings, urgent escape, ((masterpiece)), ((best quality)), 8k, high detailed, ultra-detailed, gondola, ((twisting canals)), (ancient architecture), (urgent paddling), cinematic chase.
10. Futuristic sports cars racing on a vertical loop track against a sci-fi cityscape, cars defying gravity, ((speed trails)), (dizzying heights), (spectacular crashes), the thrill of cutting-edge technology.
11. Cat rapidly zigzagging across a rooftop, avoiding swooping birds under a stormy sky.
12. A sequence of a cow performing acrobatic stunts over a series of colorful, abstract platforms that morph shapes.
13. The dog bursts through a thicket, darting from a foggy forest to a steep hillside, rocks crumbling under its paws as it charges towards a roaring river below.
14. Thundering across a vast desert plain, the elephants race over dunes and dodge sandstorms, before swiftly traversing through a rocky canyon, bounding over boulders and leaping across narrow ravines.
15. A giraffe navigating a city during a robot uprising, with quick cuts showing chaotic battles, explosions, and futuristic technology in a high-stakes escape scenario.
16. A horse leading a wild stampede across a stormy beach with waves crashing, depicted with swift, sweeping camera moves, cinematic composition.
17. Intense motorcycle escape from a volcanic eruption, with transitions from lava-filled landscapes to ash-clouded skies.
18. A futuristic robot uprising, ((lasers firing)), metallic drones, explosions, debris, ((screaming civilians)), dystopian cityscape.
19. Sheeps engaging in a high-speed pursuit through a cyberpunk city, the scene rapidly transitioning between neon-lit streets, bustling marketplaces, and towering skyscrapers.
20. A pulse-pounding sequence of a train barreling through a treacherous storm, the scene transitioning between lightning-lit skies and torrents of rain to flooded tracks and collapsing bridges.
21. A truck rushing away from a treacherous mountain pass during a blizzard, with sudden avalanches and rockslides adding to the danger.
22. A zebra sprinting across the busy lanes of Times Square in New York City, with scene transitions occurring quickly as it moves from iconic billboards to bustling sidewalks filled with tourists.

## MovieGen

1. A green monster made of plants walks through an airport.
2. A marble goes through a glass cup, breaking it into pieces.
3. A droplet of water falling onto a hot surface, instantly evaporating into a wisp of steam that swirls gracefully into the air.
4. An old man wearing a green dress and a sun hat taking a pleasant stroll in Johannesburg South Africa during a beautiful sunset.
5. A person on a hoverboard colliding with a wall, the board stopping abruptly.
6. A toy robot wearing blue jeans and a white t-shirt taking a pleasant stroll in Johannesburg South Africa during a winter storm.
7. In a marathon race, a female athlete gradually sprints ahead of the male athletes.
8. A teenager eating a slice of pizza, cheese stretching as they pull it away.
9. A man in a suit fights monsters.
10. A dog made of ice melts completely in a hot summer day.
11. A truck right alongside a flowing river, capturing the movement of the water and the surrounding forest.
12. A group of skateboarders perform tricks on ramps and rails at a skate park, showcasing their skills.
13. A hot air balloon descending back to the ground.
14. Chef chopping onions in the kitchen for the preparation of the dish.
15. Zoom in shot to the face of a young woman sitting on a bench in the middle of an empty school gym.
16. The couple runs hand in hand to release a sky lantern, then watches it drift upward into the night sky, carried by the wind with the stars shining above.
17. Aerial view shot of a cloaked figure elevating in the sky between skyscrapers.
18. A softball player sliding safely into second base.
19. A giraffe in a lifeguard outfit, sitting atop a high chair and watching over a crowded pool.
20. A speed skater accelerating during a short track race.

## H    Detailed Analysis on Reward Function for Perceptual Video Quality

Figure 10 shows that different metrics in reward functions for perceptual video quality often exhibit negative or weak correlations. For example, dynamic degree tends to be negatively correlated with many other metrics, indicating that optimizing exclusively for one metric can either reduce motion dynamics or undermine temporal consistency and aesthetic quality. These findings underscore the need to balance potentially conflicting reward functions, rather than prioritizing any single one in isolation, and emphasize the importance of a carefully calibrated approach to evaluating generated videos.

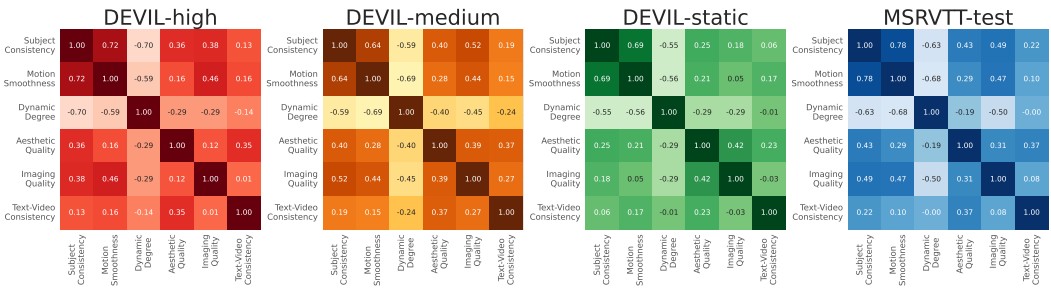

Figure 10: Correlation between reward functions for perceptual video quality.

## I    Prompt of AI Feedback

---
**Prompt for AI Feedback from VLMs**

You are a helpful assistant that evaluates the quality of a generated video from a textual prompt.

Compare the text prompt and generated video and evaluate the quality (visual quality, proper dynamics, etc...) of the video.

First explain the reasoning, then present the final assessment. Start the reasoning with 'Reasoning: '.

After explaining the reasoning, present the final assessment with 'Assessment: '.

Your final 'Assessment' should be a single-number score from 1 to 10, not as a fraction.

When evaluating, consider the following points:

    - Visual Quality: Evaluate the clearness, resolution, brightness, aesthetic appeal of the video.

    - Dynamics: Evaluate whether the video demonstrates appropriate dynamics, ensuring it avoids excessive movement in situations meant to be static or insufficient movement in situations intended to be dynamic.

    - Smoothness, Consistency, and Naturalness: Assess the smoothness, consistency, and naturalness of shape and motion for objects, animals, and humans.

    - Contents: Evaluate whether the video content aligns with the given text prompt.

Textual Prompt: {instruction}

Video: {video_file}

---

# J Further Results for Calibrating Reward to Preference Feedback

## J.1 Basic Prompts

Figure 13 and Figure 14 show the two-dimensional histogram and correlation between reward function and AI feedback from Gemini [14] and GPT-4o [13], and Figure 11 represents the coefficient of calibrated reward designed for GPT-4o. The relative weighting assigned to the dynamic degree changes according to the dynamics grade of the prompt. Specifically, prompts with a high dynamics grade, i.e., DEVIL-high, place greater weight on the dynamic degree. In contrast, prompts that describe slight motion, i.e., DEVIL-medium and DEVIL-static, place a smaller weight on it. This behavior matches the pattern observed in reward calibration with Gemini (Figure 4). GPT-4o exhibits a stronger inclination toward dynamics than Gemini.

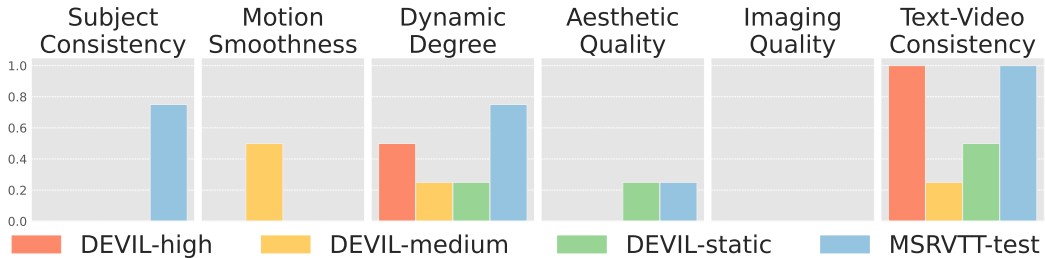

Figure 11: Coefficients of calibrated reward with GPT-4o.

## J.2 Challenging Prompts

This section describes the reward calibration procedure and results for two challenging prompt sets, DEVIL-very-high and MovieGen, which were introduced to evaluate our method with larger T2V models, such as CogVideoX [19] and Wan 2.1 [20]. Following the methodology for reward calibration with Latte (see Section 4), we generated 64 videos per prompt using Wan 2.1-1.3B [20]. Consistent with observations in Appendix J.1, using solely text-video consistency is insufficient to fully capture AI feedback from Gemini (Figure 15). We choose the combination of weights $w_i$ to maximize correlation with Gemini's evaluations. The coefficients of calibrated weights are shown in Figure 12.

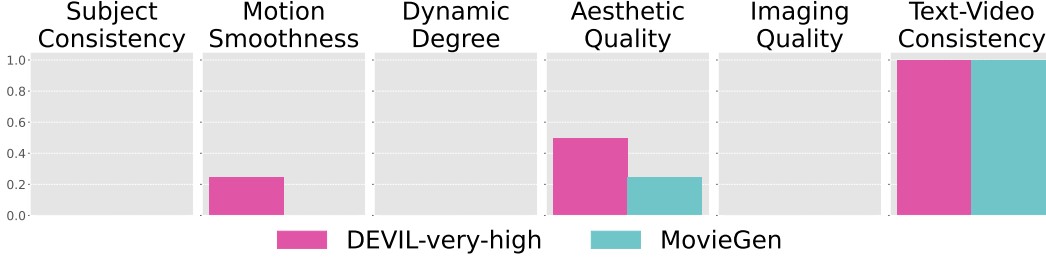

Figure 12: Coefficients of calibrated reward with Gemini.

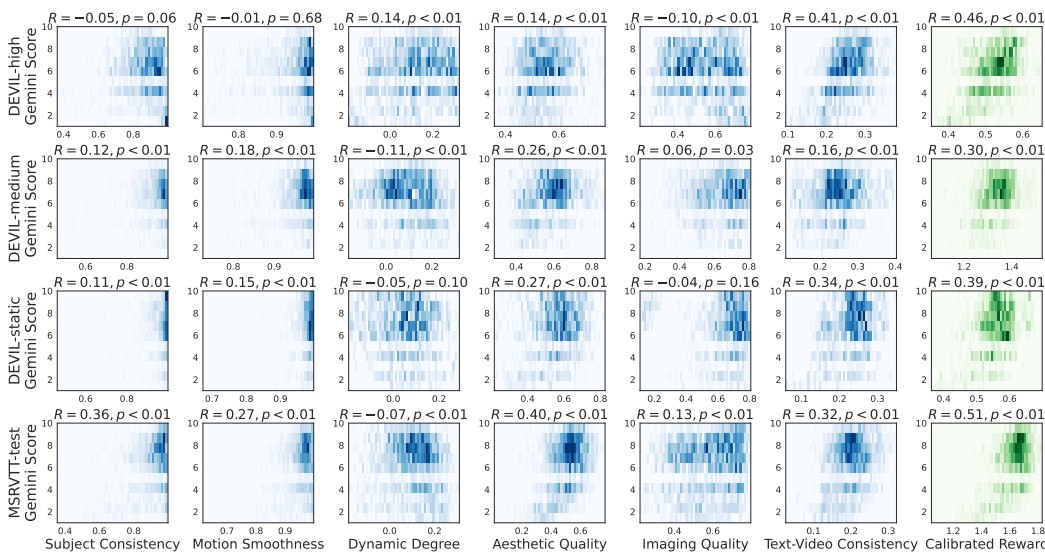

Figure 13: 2D-histogram and correlation between reward function and AI feedback from Gemini.

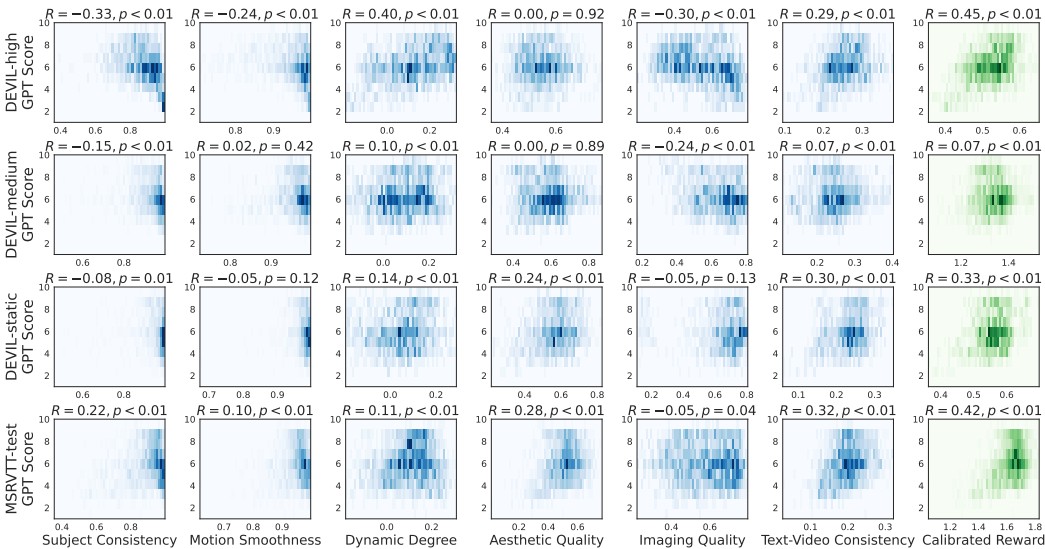

Figure 14: 2D-histogram and correlation between reward function and AI feedback from GPT-4o.

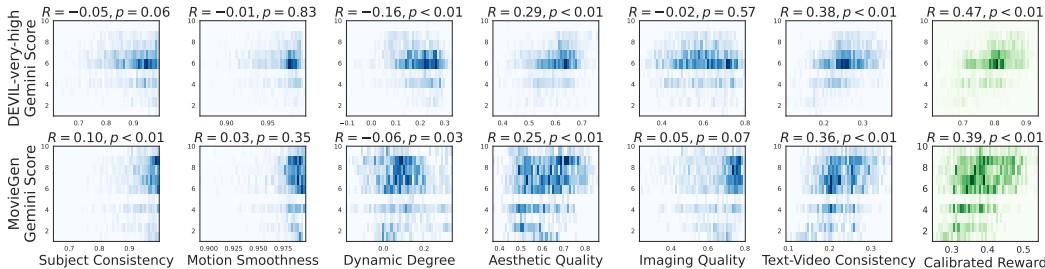

Figure 15: 2D-histogram and correlation between reward function and AI feedback from Gemini for challenging prompt sets, DEVIL-very-high and MovieGen.

## J.3 Cost of Reward Calibration

As described in Section 4.1 and Appendix J.2, we generate 64 videos per prompt using pre-trained Latte and Wan 2.1 models. Compared to naively querying VLMs at every inference step, our calibration approach is substantially more cost-efficient, since the VLM queries are amortized through a one-time weight estimation. Table 2 summarizes the difference in per-prompt query count and execution time when applying DLBS ($KB = 32$). These results demonstrate that reward calibration reduces the number of VLM queries, making large-scale search with DLBS computationally feasible.

Table 2: Comparison of query count and execution time between naive VLM queries during search and reward calibration. Assuming 15 seconds per VLM query.

| Method | Query Count | Exec. Time (sec) |
|---|---|---|
| Querying VLMs during Search ($KB = 32$) | $T{=}50 \times KB{=}32 = 1600$ | $\approx 102{,}400$ |
| Reward Calibration | **64** | $\approx \mathbf{960}$ |

## J.4 Generalization of Reward Calibration across prompts

Video generation inherently involves trade-offs between fundamental properties such as dynamics and consistency (Appendix H), which may require category-specific calibration for optimal performance. However, despite these domain-specific requirements, we hypothesize that calibrated rewards can generalize to some extent across different datasets, as they are based on shared principles of perceptual quality. To test the out-of-domain transferability, we conducted additional experiments applying the reward weights calibrated on DEVIL-high and DEVIL-medium to MSRVTT-test prompts (Table 3). We used Latte [18] as a base model and evaluated the results using VideoScore [55], a human preference-trained evaluator, measuring five key metrics along with their corresponding average scores. With the DEVIL-high reward, we can enhance other metrics while maintaining dynamics. DEVIL-medium reward, which is a closer domain to MSRVTT-test, shows a different trade-off pattern. While it slightly reduces dynamics, it significantly improves other metrics and achieves a higher average score than the MSRVTT-test reward, demonstrating higher transferability.

Table 3: Out-of-domain prompt generalization. Rewards calibrated on DEVIL-high/medium applied to MSRVTT-test prompts. All metrics are derived from VideoScore [55]. VQ = Visual Quality; TC = Temporal Consistency; DD = Dynamic Degree; T2V Align. = Text-to-video Alignment; FC = Factual Consistency.

| (R: Reward, P: Prompt) | VQ | TC | DD | T2V Align. | FC | Average |
|---|---|---|---|---|---|---|
| Latte | 2.32 | 2.01 | **2.91** | 2.67 | 2.07 | 2.40 |
| + DLBS ($KB = 8$) | | | | | | |
| with R=MSRVTT, P=MSRVTT | **2.50** | **2.27** | 2.88 | **2.74** | 2.28 | 2.53 |
| with R=DEVIL-medium, P=MSRVTT | 2.49 | 2.26 | 2.89 | 2.73 | **2.31** | **2.54** |
| with R=DEVIL-high, P=MSRVTT | 2.36 | 2.03 | 2.90 | 2.68 | 2.10 | 2.42 |

# K   Correlation between VLM and Human Evaluation

As mentioned in prior research [44–46], evaluation from VLMs such as Gemini and GPT-4o exhibits a high correlation with human assessment compared to other existing metrics. As an experiment, we measured the correlation between the AI feedback from these VLMs and human labels in the TVGE dataset [74]. As shown in Figure 16, Gemini achieved a correlation of 0.49, and GPT-4o achieved 0.51. Consequently, optimizing for these VLM rewards is a valid way to improve human-perceived quality, rather than merely "gaming" the metrics.

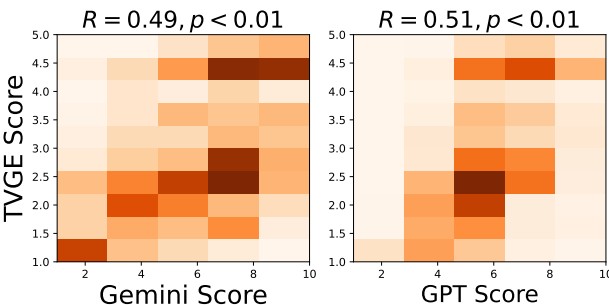

Figure 16: Correlation between VLM outputs and human labels in the TVGE dataset.

For a deeper analysis of failure cases, we qualitatively examined the top 5% outliers between human preference labels in the TVGE dataset and VLM (Gemini) evaluation (Figure 17). As far as we observed, VLM sometimes makes subtle mistakes, but we did not see any critical failures.

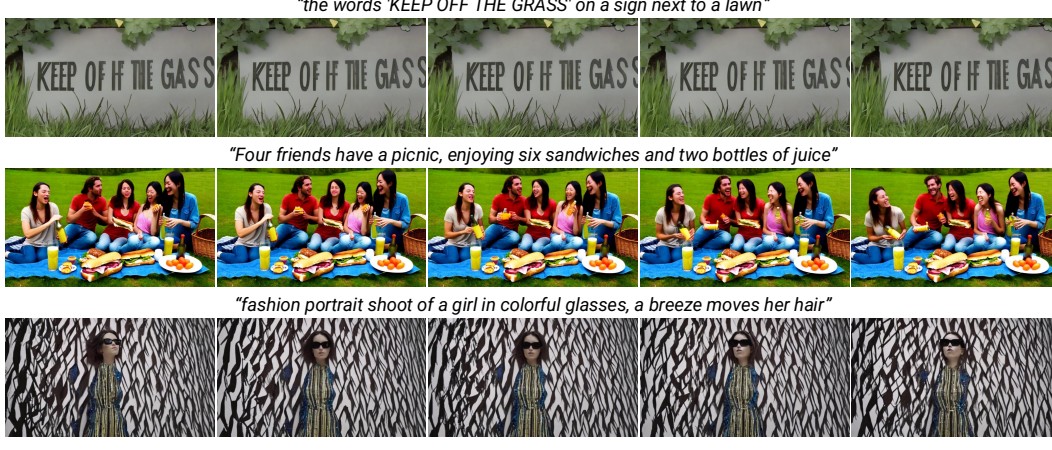

Figure 17: Misaligned cases of Gemini-based evaluation with human preferences. (**Top**) For prompts specifying text, such as *"the words 'KEEP OFF THE GRASS' on a sign next to a lawn,"* the VLM was significantly harsher than human evaluation on text rendering (VLM: 2/10, Human: 4.5/5). (**Middle**) For prompts specifying quantities, such as *"Four friends have a picnic, enjoying six sandwiches and two bottles of juice,"* the VLM was more lenient than human evaluation (VLM: 8/10, Human: 1.5/5). (**Bottom**) For *"fashion portrait shoot of a girl in colorful glasses, a breeze moves her hair,"* despite missing arms in the generated person, the VLM was misled by distracting background patterns, possibly mistaking them for curtain-like elements that obscure the arms behind the background (VLM: 8/10, Human: 1.2/5).

## L    Qualitative Evaluation of Calibrated Reward

We provide best-of-64 videos by individual rewards and VLM calibrated rewards in Figure 18. Videos selected solely on a single metric can over-optimize one aspect while neglecting others, whereas those chosen via VLM-calibrated rewards exhibit a more balanced quality. For instance, videos chosen solely based on temporal consistency (i.e., subject consistency and motion smoothness) or frame-by-frame quality (i.e., aesthetic quality, imaging quality) tend to lack dynamic movement, whereas those selected based on dynamic degree often lose temporal consistency. Evaluations relying on a single metric also fail to reflect the given prompt in some cases. Text-video consistency, which often exhibits a high correlation with VLM-based evaluation among individual metrics (Figure 3), is relatively effective in capturing the overall quality of a video. However, it may overlook certain aspects, such as frame-wise artifacts. In contrast, videos selected using VLM-calibrated rewards exhibit a more balanced overall quality.

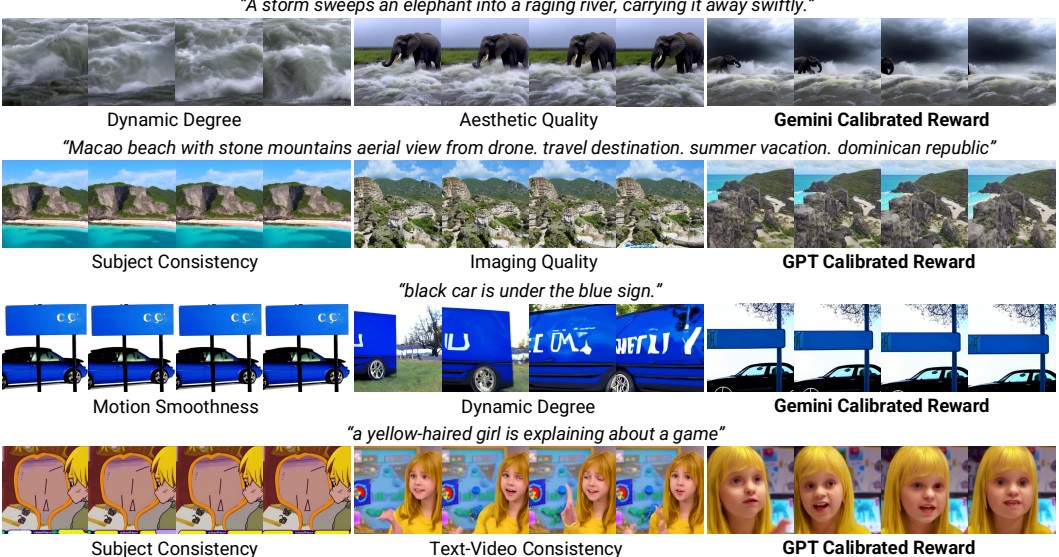

Figure 18: We select the video with the highest reward out of 64 randomly generated candidates for each prompt, drawn from DEVIL-high, DEVIL-medium, DEVIL-static, and MSRVTT-test (arranged from top to bottom). Videos chosen using VLM-calibrated rewards achieve a more balanced quality compared to those relying on any single metric. For instance, when subject consistency, motion smoothness, or aesthetic quality serves as the sole selection criterion, the resulting videos often lack dynamic movement, whereas prioritizing dynamic degree can compromise temporal consistency. Moreover, single-metric evaluations may occasionally fail to align with the intended prompt.

## M  Further Results for Diffusion Latent Beam Search

### M.1  Scaling Search Budget with GPT-4o Calibrated Reward

We measure the performance using a reward calibrated to GPT-4o (Figure 19). DLBS improves all the calibrated rewards the best as the search budget $KB$ increases (especially $KB = 16, 32$), while BoN and GS, in some cases, eventually slow down or saturate the performance. Notably, an LA estimator with a small search budget ($KB = 8, T' = 6$) is comparable to or even outperforms DLBS ($KB = 32$).

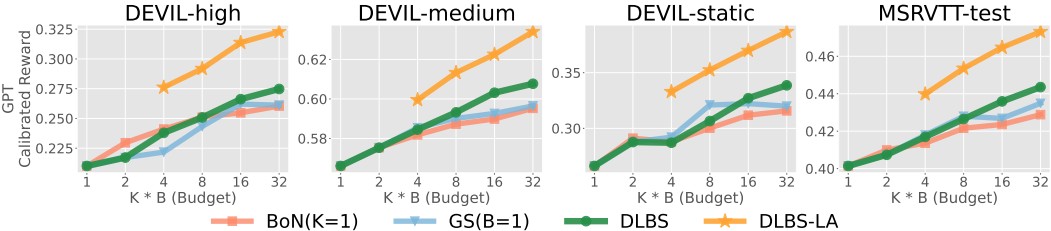

Figure 19: Inference-time search on reward calibrated to GPT-4o.

### M.2  Scaling Search Budget to Larger Regimes

Figure 20 and Figure 21 show the performance of inference-time search on DEVIL-medium and MSRVTT-test that includes the results with $KB = 64$. We can observe that the increasing trends still continue.

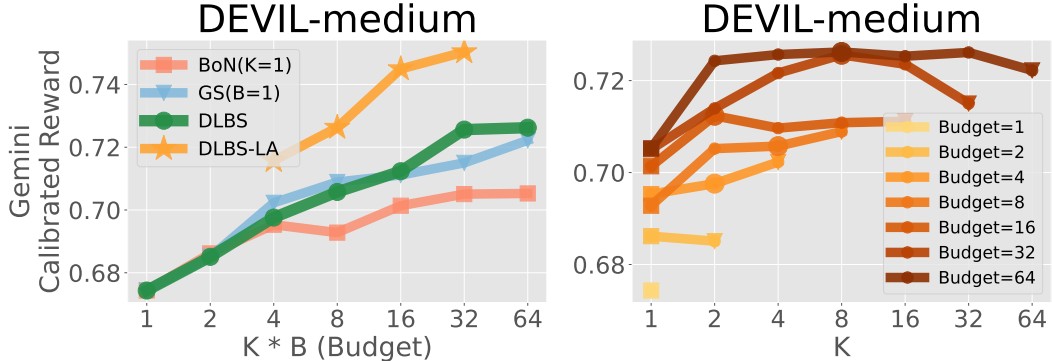

Figure 20: Inference-time search on reward calibrated to Gemini including $KB = 64$.

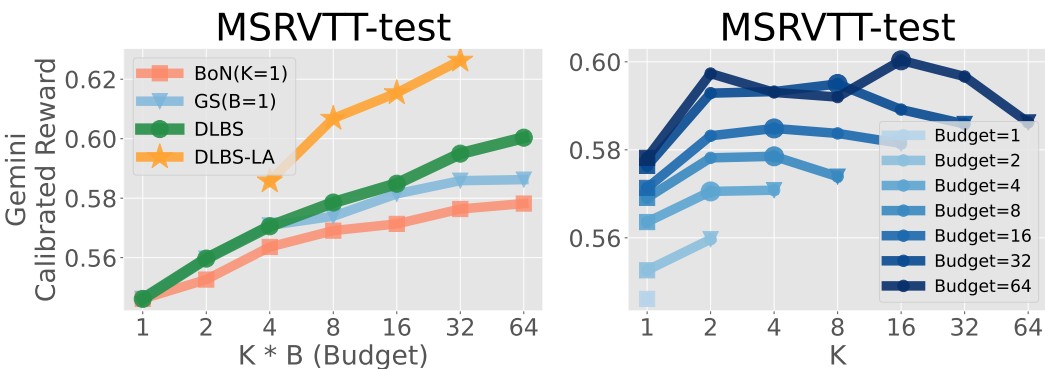

Figure 21: Inference-time search on reward calibrated to Gemini including $KB = 64$.

## M.3 Full Results for Scaling Trend of DLBS

Figure 22 demonstrates the scaling trend of DLBS, proportional to the search budget, under various choices of $K$. The results show that there is an optimal balance between the number of latent $K$ and the number of beams $B$ under the same budget. For instance, as we increase the budget to $KB = 16, 32$, we have a peak around $K = 4, 8, 16$, which is about 25–50% of the budget. This implies that balancing possession and exploration of diffusion latents in DLBS helps search for the best outputs robustly.

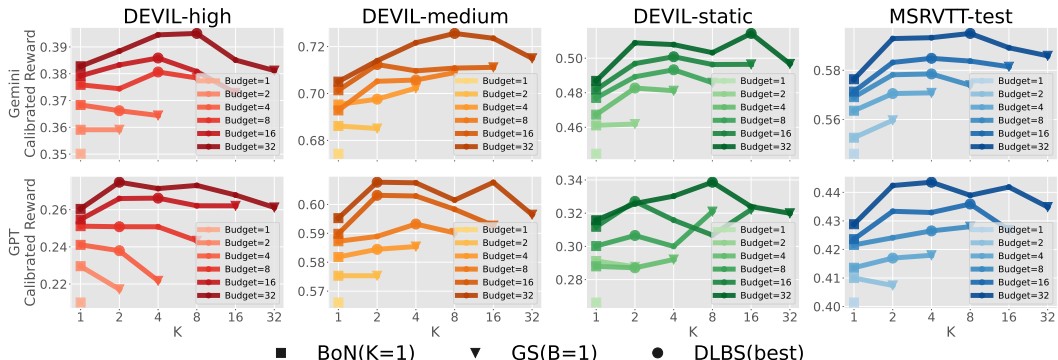

Figure 22: DLBS can improve the performance in any prompts or reward, as we increase the search budget $KB \in \{1, 2, 4, 8, 16, 32\}$. In addition, we can see an optimal balance between the number of latent $K$ and the number of beam $B$ under the same budget. For instance, as we increase the budget to $KB = 16, 32$, we have a peak around $K = 4, 8, 16$, which is about 25–50% of the budget.

## M.4 Further Analysis on Lookahead Estimator

Figure 23 (**Left**) demonstrates that increasing the number of reward estimation steps $T'$ in the LA estimator leads to improved reward prediction performance for $\mathbf{z}_t$ during the denoising process. This finding suggests that extending the LA steps enables a more effective search based on accurate reward predictions, particularly in the early stages of the denoising. As shown in Figure 23 (**Right**), enlarging the look-ahead horizon increases the reward gain to $T' = 6$; beyond this point, e.g., $T' = 20$ offers no significant benefit while multiplying the computational cost. Accordingly, we fix $T' = 6$ in the main experiments, as it captures nearly all the attainable gains at minimal cost. These results were obtained on Latte [18] using a DDIM sampler.

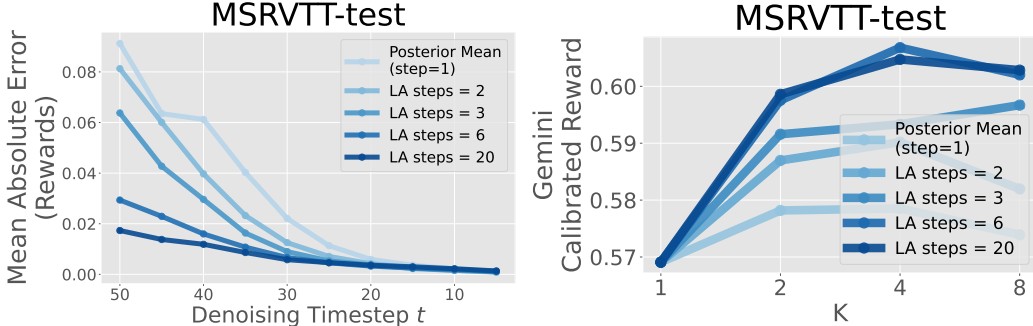

Figure 23: (**Left**) Comparison of the reward estimation error for different LA steps $T'$. We evaluate the reward predicted by the LA estimator, which projects $\mathbf{z}_t$ to $\tilde{\mathbf{z}}_{0|\tilde{t}(0)}$ in $T'$ steps (Algorithm 2) and computes $r'(\tilde{\mathbf{z}}_{0|\tilde{t}(0)})$, against the actual reward obtained by projecting $\mathbf{z}_t$ to $\mathbf{z}_0$ in $t$ steps using a DDIM sampler ($\eta = 1.0$) and evaluating $r'(\mathbf{z}_0)$. (**Right**) Impact of $T'$ on search performance. Reward improves rapidly up to $T' = 6$ but saturates thereafter; using $T' = 20$ offers no measurable gain. These results show that a modest $T'$ is sufficient in practice.

To verify that this behavior is not specific to DDIM sampler, we conducted the same ablation with an SDE-DPMSolver++ [56] on Wan 2.1-1.3B [20] (Figure 24). Note that the notation follows Algorithm 4. Specifically, $M'$ in the SDE-DPMSolver++ setting corresponds to the $T'$ used with the DDIM sampler. We observed the same pattern shown in Figure 23. Increasing the look-ahead horizon $M'$ monotonically improves the LA estimator's reward prediction. Search reward gain up to roughly $M' = 6$, after which gains saturate. For example, $M' = 12$ yields no measurable improvement while incurring substantially higher cost. This cross-sampler and cross-model consistency provides a practical guideline for choosing $M'$: a modest horizon ($\approx 6$) captures nearly all attainable benefit.

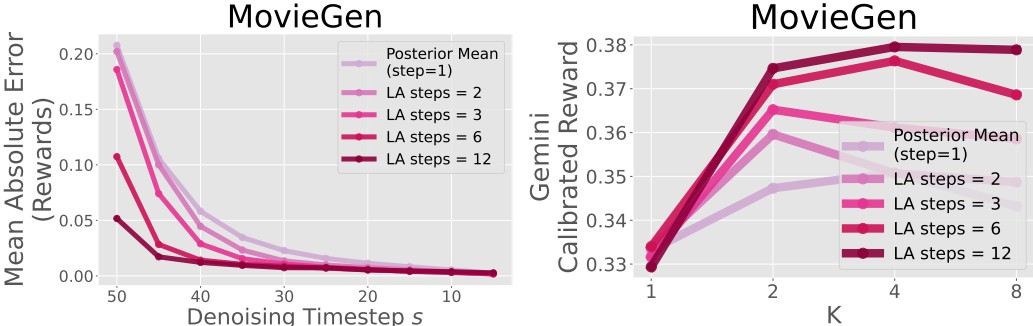

Figure 24: (**Left**) Comparison of the reward estimation error for different LA steps $M'$. We evaluate the reward predicted by the LA estimator, which projects $\mathbf{z}_{t_s}$ to $\tilde{\mathbf{z}}_{t_M | t_{\tilde{s}(0)}}$ in $M'$ steps (Algorithm 4) and computes $r'(\tilde{\mathbf{z}}_{t_M | t_{\tilde{s}(0)}})$, against the actual reward obtained by projecting $\mathbf{z}_{t_s}$ to $\mathbf{z}_{t_M}$ in $(M - s)$ steps using a SDE-DPMSolver++ [56] and evaluating $r'(\mathbf{z}_M)$. (**Right**) Ablation study of $M'$ on search performance with SDE-DPMSolver++ [56] on Wan 2.1-1.3B [20]. Reward improves rapidly up to $M' = 6$ but saturates thereafter; using $M' = 12$ offers no measurable gain. These results show that a modest $M'$ is sufficient in practice.

## M.5    Ablation Study for Diffusion Steps

**Scaling Diffusion Steps**  Figure 25 (**Left**) shows the performance when increasing the number of denoising steps $T$. Since DDIM exhibits fast convergence [42], BoN with a larger $T$ does not improve the reward much. DLBS improves performance when scaling denoising steps to $T = 200$ more than BoN, which implies that DLBS benefits from larger computational resources in denoising. However, as Figure 6 (**Left**) indicates, these gains are smaller than those obtained by widening the beam budget $KB$ or leveraging the LA estimator.

**Range of Diffusion Steps for Search**  We investigate which range of diffusion steps DLBS should be applied to for effective search. Unlike Kim et al. [15], which applies GS only during the initial 5–10 steps, our results in Figure 25 (**Right**) show that applying DLBS throughout all steps leads to substantially better performance. This suggests that applying DLBS entirely is more effective than focusing on the early stage.

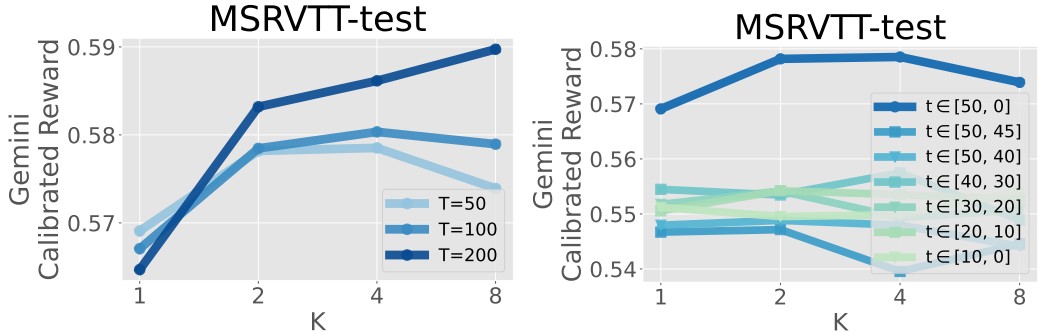

Figure 25: (**Left**) Scaling the denoising steps $T$. (**Right**) Range of denoising steps $t \in [50, 0]$ to apply search methods. While Kim et al. [15] applies GS in the first 5–10 steps, DLBS over the entire diffusion steps yields the largest improvement.

## M.6    Ablation Study for $\eta$ in DDIM scheduler

Figure 26 illustrates how varying the value of $\eta$ in DDIM influences search performance. Here, $\eta$ controls the degree of randomness in the DDIM scheduler: $\eta = 0.0$ corresponds to the deterministic version of DDIM, while $\eta = 1.0$ is equivalent to DDPM. As $\eta$ decreases below $1.0$, performance in terms of the final reward diminishes, presumably because lowering the randomness in the sampling process narrows the scope of exploration.

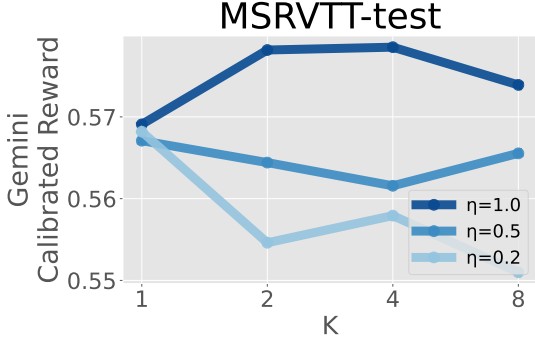

Figure 26: Comparison among different $\eta$ in DDIM sampler.

## M.7   Comparison with Gradient-Based Search Methods

**Applicability to diverse reward models**   One advantage of our zero-order search framework is its applicability to reward models for which computing gradients is computationally prohibitive (e.g., large-scale VLMs) or fundamentally impossible (e.g., human evaluators or external API-based models). To demonstrate this, we simulated search with human feedback by employing VideoScore [55], a VLM-based reward model trained on human evaluations, as the reward function. As shown in Table 4, DLBS-LA with VideoScore as the evaluator achieved substantial improvements over the vanilla baseline, suggesting that high-performance VLMs or human evaluators can, in principle, be directly incorporated as reward functions in our framework.

Table 4: Results of DLBS-LA with VideoScore as the evaluator, illustrating its applicability to reward models for which computing gradients is computationally prohibitive (e.g., large-scale VLMs) or fundamentally impossible (e.g., human evaluators or external API-based models).

| Method | VideoScore |
|---|---|
| Latte | 2.40 |
| + DLBS-LA ($KB = 4$, $T' = 6$) with VideoScore | **2.69** |

**Efficiency in time and memory**   We further compared DLBS with DAS [61], a first-order gradient-based method based on Sequential Monte Carlo. Experiments were conducted on Stable Diffusion 1.5 [21] with LAION Aesthetic V2 [50] as the reward model, using an NVIDIA RTX 6000 Ada (48GB). Table 5 shows that under the assumption of equal execution time, DLBS (a zero-order method) takes the lead because it can have a larger search budget, which refers to the number of particles used for search, i.e., $KB$ for DLBS and $N$ for DAS. Our observation that a zero-order method achieves better performance than a first-order method under the equal execution time aligns with prior findings on inference-time search for image generation [41].

Gradient-based search methods also exhibit significant increases in memory usage due to gradient computations required for the reward function and the VAE decoder. In other words, gradient-based methods are actually inefficient in terms of memory cost. The results shown in Table 5 are based on a single evaluator and a single frame (i.e., image generation). Note that for video generation, as mentioned in Figure 3, a single evaluator metric does not correlate well with perceptual quality, necessitating the combination of multiple evaluators, which roughly multiplies the memory requirements for gradient calculation. Additionally, video generation models do not decode just one frame from the VAE (maximum frames are 81 for Wan 2.1 [20] and 49 for CogVideoX [19]). The gradient increase would be significantly larger in video generation than in image generation, making gradient-based methods almost impossible in practice. For reference, the memory usage of vanilla Latte, DLBS-LA, and DAS in Table 6.

Table 5: Comparison between DLBS and DAS on Stable Diffusion 1.5 with LAION Aesthetic V2 reward. DLBS (a zero-order method) achieves better performance than DAS (a first-order method) under equal execution time.

| Method | Score | Time (sec) | Memory (GB) |
|---|---|---|---|
| SD 1.5 | 5.81 | 2 | 4.3 |
| + DAS ($N = 8$) | 6.59 | 108 | 14.2 |
| + DLBS ($K = 8, B = 2$) | **6.63** | **103** | **5.9** |
| + DAS ($N = 16$) | 6.68 | 220 | 14.2 |
| + DLBS ($K = 16, B = 2$) | **6.69** | **209** | **5.9** |

Table 6: Memory usage of search methods on Latte. The memory advantage of DLBS-LA (a zero-order method) becomes more critical in video generation.

| Method | Latte | + DLBS-LA | + DAS |
|---|---|---|---|
| Memory Usage (GB) | 16.3 | 38.9 | >48.0 (**OOM**) |

## M.8 Comparison with Other Inference-Time Search Methods

Concurrently with our work, inference-time search based on zero-order sequential Monte Carlo (SMC) has been proposed. We include a comparison with FK Steering [41] in Figure 27, where the resampling mechanism in the SMC-based methods does not occur frequently enough, preventing them from surpassing BoN performance, in our text-to-video experiments.

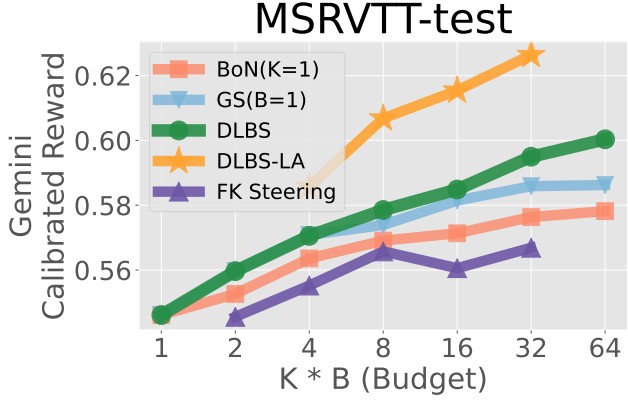

Figure 27: Comparison with FK steering [41].

## M.9 Scalability to Long Videos

To demonstrate that our method is scalable even when extending the frame count to the model's maximum, we conducted experiments with the maximum frames for CogVideoX-5B (49 frames, 6 seconds) and Wan 2.1-1.3B (81 frames, 5 seconds). The reward, which was calibrated using 33-frame, 2-second videos generated by Wan 2.1-1.3B, was applied as-is. As shown in Table 7 and Table 8, we confirmed that even with longer frames, the reward values could be improved more efficiently than the BoN baseline.

Table 7: Scalability results on CogVideoX-5B with 49 frames, 6 seconds using DEVIL-very-high prompts.

| Method | $KB$ | Reward | Inference Compute (NFE) |
|---|---|---|---|
| CogVideoX-5B | 1 | 0.429 | 50 |
| + BoN | 64 | 0.474 | 3200 |
|  | 128 | 0.478 | 6400 |
| + DLBS | 16 | 0.481 | 1200 |
|  | 32 | 0.490 | 2400 |
| + DLBS-LA | 8 | **0.497** | 2500 |
|  | 16 | **0.517** | 4900 |

Table 8: Scalability results on Wan 2.1-1.3B with 81 frames, 5 seconds using MovieGen prompts.

| Method | $KB$ | Reward | Inference Compute (NFE) |
|---|---|---|---|
| Wan 2.1-1.3B | 1 | 0.313 | 50 |
| + BoN | 128 | 0.357 | 6400 |
|  | 256 | 0.360 | 12800 |
| + DLBS | 16 | 0.357 | 1200 |
|  | 32 | 0.371 | 2400 |
| + DLBS-LA | 8 | **0.373** | 2500 |
|  | 16 | **0.393** | 5000 |

## M.10 DLBS with Larger Text-to-Video Models

We have tested our method on VideoCrafter2 [3] (1.9B) and CogVideoX-5B, 2B [19] and Wan 2.1-14B, 1.3B [20]. Our experiments confirm that our DLBS and DLBS-LA yield significant improvements, indicating their effectiveness can be observed in larger video generation models.

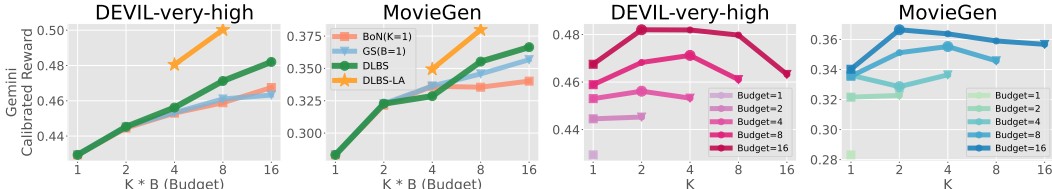

Figure 28: Inference-time search with CogVideoX-5B [19].

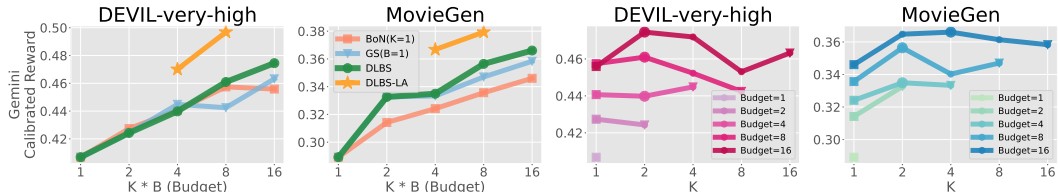

Figure 29: Inference-time search with CogVideoX-2B [19].

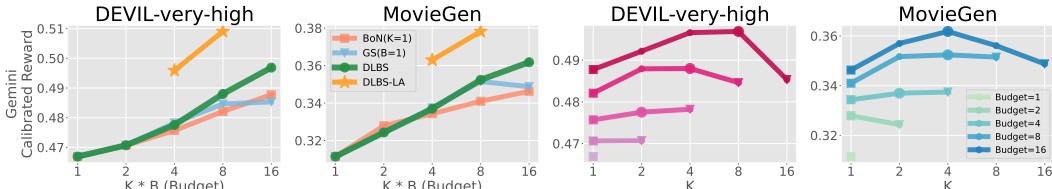

Figure 30: Inference-time search with Wan 2.1-14B [20].

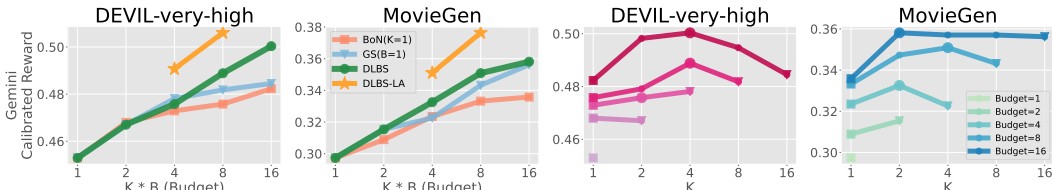

Figure 31: Inference-time search with Wan 2.1-1.3B [20].

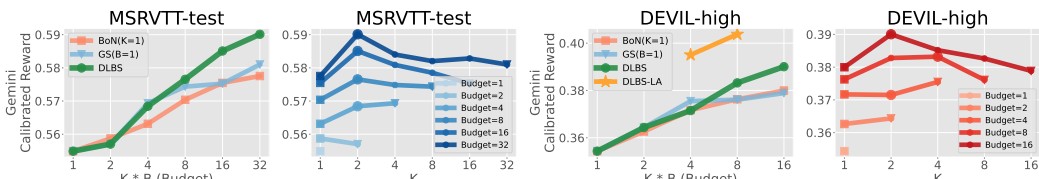

Figure 32: Search with VideoCrafter2 [3].      Figure 33: Search with CogVideoX-5B [19].

## M.11 Further Results in AI and Human Evaluation

We show full results of evaluations using VideoScore [55], a metric trained on human judgments that evaluates videos at 8 fps across five dimensions and outputs scores ranging from 1.0 to 4.0 (see Figure 34). Under this metric, videos generated with our DLBS consistently outperformed those generated without search and with BoN.

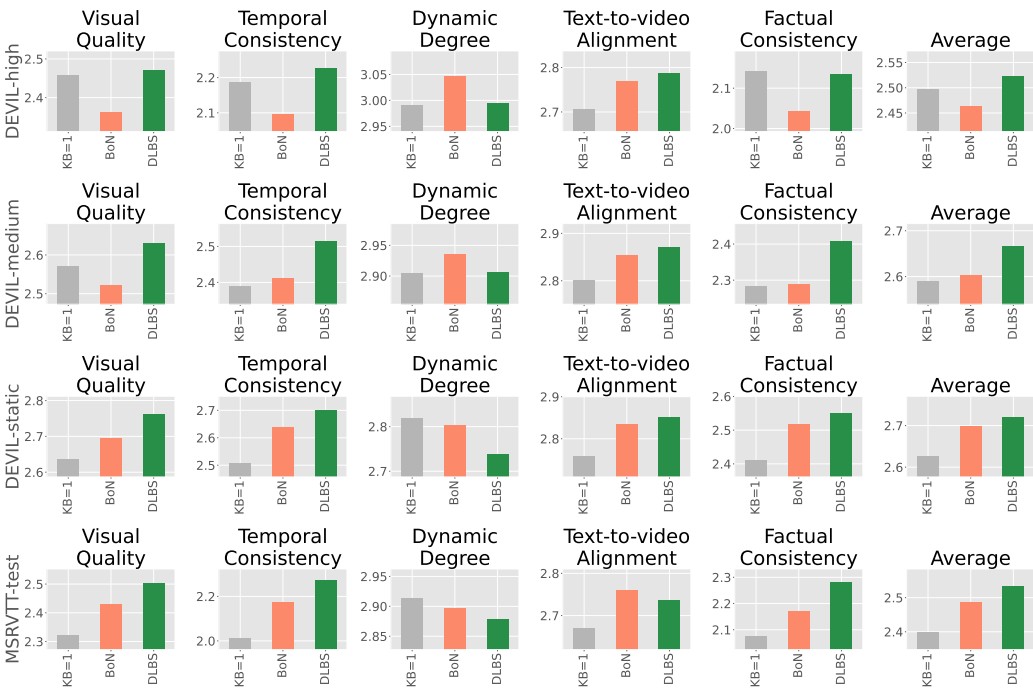

Figure 34: DLBS on calibrated reward also improves another qualitative metric, the most, VideoScore [55], which is not involved in a reward calibration.

We also show additional results of human judgment by three human evaluators (see Figure 35). These experiments confirmed that, regarding human preference (win rate), content generated with our search strategy consistently outperformed content produced without search.

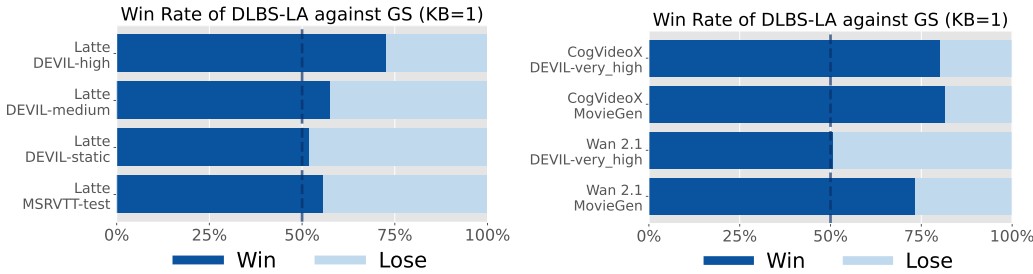

Figure 35: Human evaluation results. We searched videos using the Gemini calibrated reward and asked three human evaluators to compare outputs from GS ($KB = 1$) and DLBS-LA ($KB = 8, T' = 6$). "Win" indicates that the video generated by DLBS-LA was preferred.

## M.12 Qualitative Results for DLBS

Qualitative results are shown in `https://sites.google.com/view/t2v-dlbs`.

## M.13 DLBS for Image Generation

We adopt PixArt-$\alpha$ [69] as our base text-to-image generation model. For evaluation, we directly reuse the prompt set of 45 common animal categories from prior works [75, 63]. As a reward model, we employ the LAION aesthetic predictor [50] to assess image quality.

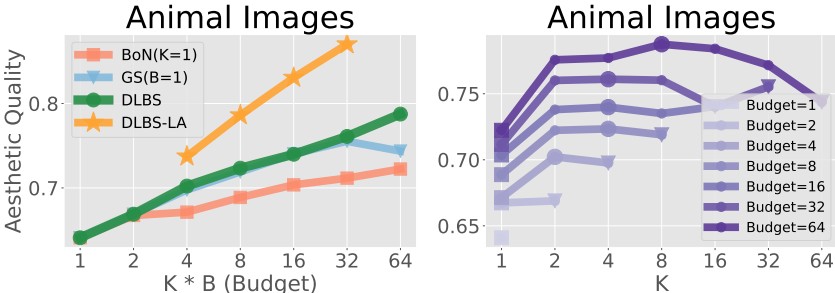

Figure 36: Inference-time search with PixArt-$\alpha$ [69].

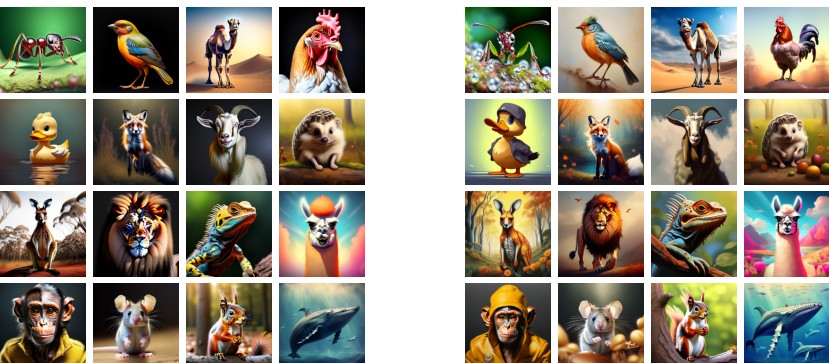

Figure 37: Qualitative Results in inference-time search with PixArt$-\alpha$ [69]. **(Left)** GS ($KB = 1$). **(Right)** DLBS-LA ($KB = 32, T' = 6$).

# N   Extended Related Works

**Aligning Diffusion Models via Finetuning**   Alignment by finetuning text-conditioned models has been investigated for image [10] and video [46, 45] generation. Typically, LoRA [76] in a backbone model [77] is finetuned through policy gradient [75, 78, 79], direct preference optimization [80, 44, 81, 38, 82], reward-weighted regression [83], or direct reward gradient [84, 85, 45, 86]. Some train an extra model for better initial noise space [87–89]. In contrast, we focus on the search over the denoising process at inference time, which does not require any model updates and may not degrade the original performance.

**Evaluation of Text-to-Video Generation**   While there are several conventional metrics for video generation (or the one repurposed from image generation) such as SSIM [90], IS [91], LPIPS [92], or FVD [93], those are not always suitable to evaluate how the quality of contents in video is, which is much more emphasized in text-to-video generation [74]. It has been a long-standing challenge to comprehensively and semantically evaluate the dynamics of contents or physical commonsense in generated videos [9, 53]. To deal with that, VBench [12] has recently been proposed as a suite of holistic evaluations for text-to-video generation to reflect the perceptual aspect of the quality, such as consistency, smoothness, aesthetics of contents, or text–video alignment. Moreover, inspired by the success in LLMs [94–96], we could leverage VLMs, which become more capable these days, as a proxy of human evaluation of the contents [45]; by finetuning CLIP-based models [55, 71, 97], or prompting GPT-4o [13] or Gemini [14]. Our paper adopts AI feedback from VLMs as an alternative to human rater, and proposes a recipe to calibrate a reward to other sources of feedback (such as AI or human feedback), by considering a linear combination of fine-grained metrics.

