# OpenReview forum: "Inference-Time Text-to-Video Alignment with Diffusion Latent Beam Search"
_NeurIPS.cc/2025/Conference — NeurIPS 2025 poster_

### Official Review · Reviewer_3kkj · 2025-07-02

**Clarity:** 3
**Significance:** 2
**Originality:** 3
**Rating:** 4
**Confidence:** 4

**Summary:**

This paper presents Diffusion Latent Beam Search (DLBS), a gradient-free inference-time search algorithm for aligning text-to-video diffusion model outputs with perceptual quality and prompt intent. The method is developed based on the beam search, which can select a better diffusion latent to maximize a given alignment reward at inference time. To address the noisy evaluation of intermediate latents, the authors propose a lookahead estimator using deterministic DDIM steps to better approximate the final output. Additionally, the paper highlights a significant issue in current evaluation: existing perceptual metrics (e.g., smoothness, consistency) correlate poorly with human or VLM preferences. To solve this, the authors introduce a reward calibration step, learning weighted combinations of existing metrics to better match Gemini/GPT feedback.

**Questions:**

1. Can the calibrated reward weights learned on DEVIL-high be applied to other datasets (e.g., MSRVTT) with minimal degradation? Or must each dataset have its own calibration?
2. How does DLBS perform when applied to longer or higher-resolution video generation tasks (e.g., >8s or >512×512)?
3. Does DLBS reduce diversity due to beam pruning toward high-reward paths? Could you include diversity metrics to evaluate this issue?

**Ethical Concerns:**

["NO or VERY MINOR ethics concerns only"]

**Final Justification:**

The additional experiments and clarifications are appreciated and have addressed most of my concerns.

However, I still have concerns about the sensitivity of performance to the reward weight configuration. While the authors show some transferability, the paper still lacks a clear strategy or guideline on how to select or adapt reward weights in practice.

Considering the overall quality of the paper, I will maintain my positive score.

**Limitations:**

The authors discuss some limitations in Section 7.

**Quality:**

3

**Strengths And Weaknesses:**

Strengths:
* The proposed DLBS requires no model retraining, is plug-and-play, and integrates well with any existing text-to-video diffusion model.
* The recognition that no single perceptual metric aligns well with VLM or human feedback is important and convincingly demonstrated. The calibrated reward design adds real value.
* The method is benchmarked across datasets (MSRVTT, DEVIL, MovieGen), prompt difficulty levels (static, medium, dynamic), and model scales (Latte, CogVideoX, Wan 2.1). Includes both automatic and human feedback.

Weaknesses:
* While DLBS integrates known components (beam search, lookahead via DDIM), the method is more of a well-engineered combination. Its novelty lies more in system design and reward calibration.
* The improvements hinge significantly on the quality of the calibrated reward. Yet, the brute-force search for weights, though effective, may not generalize well or be feasible without VLM feedback access.
* Although using VLMs (Gemini, GPT-4o) is practical, it inherits their limitations. The assumption that VLM preferences fully reflect human preference is only partially validated. Human evaluations are relatively limited in scope.
* While DLBS improves reward-aligned outputs, its impact on sample diversity is under-explored. Could it lead to mode collapse or reduced creativity?
* The learned weight combinations for calibrated rewards differ across datasets. It’s unclear how sensitive performance is to these weights, and whether transfer across domains is viable.

---

> ### Author Rebuttal · Authors · 2025-07-31
>
> We thank the reviewer for careful reading and detailed feedback.
>
> **> W1**
>
> > While DLBS integrates known components (beam search, lookahead via DDIM), the method is more of a well-engineered combination.
>
> We believe that our contributions and implications go beyond a simple combination of existing components; beam search and lookahead via DDIM.
>
> First, we provide a unified view of existing inference-time alignment methods through several practical approximations of optimal drift $\mathbf{u}^∗(\mathbf{ν}_t, t)$ (**Section 3.1**). To mitigate these approximation errors, we introduce DLBS with an integrated lookahead estimator (**Section 3.2**).
>
> Next, we provide theoretical (**Appendix E**) and empirical (**Appendix L.4**)  analysis of approximation errors for the lookahead estimator, demonstrating that using a few steps (e.g., 6 steps) rather than traditional single-step lookahead (i.e., Tweedie’s formula) can reduce reward estimation errors.
>
> Moreover, through a comprehensive comparison of budget allocation, we show that lookahead achieves larger performance gains from search than increasing beam search budget or diffusion denoising steps (**Section 5.1**).
>
> Lastly, we demonstrate that the lookahead estimator is not limited to DDIM, but can be applied to other samplers as well—for example, SDE-DPMSolver++ used with Wan 2.1 (**Appendix D**).
>
> **> W2**
>
> > The improvements hinge significantly on the quality of the calibrated reward. The brute-force search for weights, though effective, may not generalize well or be feasible without VLM feedback access.
>
> - "not generalize well"
>
>     First, we have demonstrated generalization across models and schedulers.
>     We confirmed that rewards collected from Wan 2.1 1.3B with DPMSolver++ can improve human evaluation scores for CogVideoX 5B with DDIM solver (**Section 5.3**).
>
>     Next, we have also confirmed that for in-domain prompts, using the same reward can improve performance on unseen prompts (please see **Figure 8; Right**).
>
>     Lastly, for out-of-domain prompts and video length generalization, please see **Q1** and **Q2**, respectively.
>
> - "be feasible without VLM feedback access"
> As explained in **Section 4**, while our current approach uses VLM feedback for reward calibration, this is primarily for practical demands rather than a fundamental necessity.
>
>     Human evaluation remains the gold standard for video quality assessment, but collecting human feedback at the required scale for search is prohibitively expensive and time-consuming.
>
>     A practical approach to reduce time and cost is leveraging AI feedback from VLMs, which has shown modest alignment with human judgment on video quality (**Appendix J**). Since alignment via inference-time search requires massive reward evaluation queries, we need tractable proxy rewards that avoid reliance on humans or external VLM APIs for deployment. Existing lightweight metrics for video naturalness (e.g., `Subject Consistency`, `Dynamic Degree`) often fail to correlate with human/VLM evaluations (**Figure 3**), motivating our reward calibration approach that weights existing metrics using VLMs. While human labels would be ideal for calibration, collecting the extensive metric evaluations required would be even more costly and time-consuming than gathering preference labels from VLM APIs.
>     || Human | VLM | Lightweight Metrics |
>     |---|---|---|---|
>     | Alignment with Human Eval. | ◎ | ○ | △ |
>     | Cost | △ | ○ | ◎ |
>     | Exec. Speed | △ | ○ | ◎ |
>
>     Crucially, to demonstrate the feasibility of reward calibration without VLM APIs, we performed calibration on MSRVTT-test prompts using VideoScore (a reward model trained on human evaluations) and conducted search accordingly, showing that our approach can work in principle without external VLM dependencies.
>     | Method | VideoScore |
>     |---|---|
>     | Vanilla | 2.40 |
>     | DLBS-LA ($KB=4$, $T’=6$) + VideoScore Calibrated Reward | **2.60** |
>
> **> W3**
>
> > The assumption that VLM preferences fully reflect human preference is only partially validated. Human evaluations are relatively limited in scope.
>
> Most importantly, **Section 5.2** demonstrates that search using VLM-calibrated rewards successfully improves Human Preference and Human Preference-trained Reward Models (VideoScore) across multiple prompt sets and models.
> This supports the effectiveness of using VLM evaluation as a substitute for human evaluation in improving human evaluation outcomes.
>
> To strengthen this argument, we conducted human evaluations comparing DLBS-LA ($KB=8$, $T’=6$, $NFE=2500$) and BoN ($KB=64$, $NFE=3200$).
> While the calibrated rewards are proxy functions for human evaluation with limited responsiveness, DLBS-LA outperforms BoN in human evaluation despite requiring fewer NFEs.
>
> | Model | Prompt Set | Win Rate of DLBS-LA against BoN | Reward (DLBS-LA) | Reward (BoN) |
> |---|---|---|---|---|
> | Latte | DEVIL-medium | **69.70%** | 0.726 | 0.705 |
> | Latte | MSRVTT-test | **65.56%** | 0.607 | 0.578 |
> | CogVideoX-5B | DEVIL-very-high | **69.70%** | 0.500 | 0.476 |
> | Wan 2.1 14B | MovieGen | **60.00%** | 0.378 | 0.357 |
>
>
> **> Q1 & W5**
>
> > Can the calibrated reward weights learned on DEVIL-high be applied to other datasets (e.g., MSRVTT) with minimal degradation?
>
> To address concerns about reward transferability, we first note that video generation inherently involves trade-offs between fundamental properties such as dynamics and consistency (**Appendix G**), which may require category-specific calibration for optimal performance.
> However, despite these domain-specific requirements, we hypothesize that calibrated rewards can generalize to some extent across different datasets, as they are based on shared principles of perceptual quality.
> To test this generalization capability, we conducted additional experiments applying the calibrated reward weights calibrated on DEVIL-high and DEVIL-medium to MSRVTT-test prompts.
> We evaluated the results using VideoScore, a human preference-trained evaluator, measuring five key metrics along with their average scores.
> | (R: Reward, P: Prompt)| Visual Quality | Temporal Consistency | Dynamic Degree | T2V Alignment | Factual Consistency | Average |
> |---|---|---|---|---|---|---|
> | **Vanilla** | 2.32 | 2.01 | **2.91** | 2.67 | 2.07 | 2.40 |
> | **DLBS (KB=8)** | | | | | | |
> | + R=MSRVTT, P=MSRVTT | **2.50** | **2.27** | 2.88 | **2.74** | 2.28 | 2.53 |
> | + R=DEVIL-med, P=MSRVTT | 2.49 | 2.26 | 2.89 | 2.73 | **2.31** | **2.54** |
> | + R=DEVIL-high, P=MSRVTT | 2.36 | 2.03 | ***2.90*** | 2.68 | 2.10 | 2.42 |
>
> With the DEVIL-high reward, we can enhance other metrics while maintaining dynamics.
> DEVIL-medium reward, which is a closer domain to MSRVTT-test, shows a different trade-off pattern: while it slightly reduces dynamics, it significantly improves other metrics and achieves a higher average score than the MSRVTT-test reward, demonstrating higher transferability.
>
> Moreover, although we fixed the weight of rewards per category in the paper,  we demonstrate that expanding prompt coverage enables more flexible reward design as in **W3 from Reviewer 3Psp**.
>
> We believe that, while our reward calibration is designed for specialization to specific prompt sets, it can also achieve generalization to some extent. In particular, we can also do reward calibration against all the prompt sets jointly, which yields a good reward that works.
>
> **> Q2**
>
> > How does DLBS perform when applied to longer or higher-resolution video generation tasks (e.g., >8s or >512×512)?
>
> - Longer video generation
>
>     Our method is scalable even when extending the frame length to the model's maximum. First, note that the maximum frame length and duration of open-source SoTA video DiT models are: Wan 2.1: 81 frames, 5s (16fps), CogVideoX: 49 frames, 6s (8fps), Hunyan Video [1]: 129 frames, 5s (24fps). Therefore, experiments with >8s duration are difficult to conduct.
>
>     To demonstrate that our method is scalable even when extending the frame count to the model's maximum, we conducted additional experiments with the maximum frames for Wan 2.1 1.3B and CogVideoX 5B.
>     The reward, which was calibrated using 33 frames, 2s videos generated by Wan 2.1 1.3B, was applied as-is.
>     As a result, we confirmed that even with longer frames, the reward values could be improved more efficiently than the BoN baseline.
>
>     - CogVideoX 5B, 49 frames (6s), DEVIL-very-high prompts
>     | Method | $KB$ | Reward | Inference Compute (NFE) |
>     |---|---|---|---|
>     | Vanilla | 1 | 0.429 | 50 |
>     | BoN | 64 | 0.474 | 3200 |
>     | BoN | 128 | 0.478 | 6400 |
>     | DLBS | 16 | 0.481 | 1200 |
>     | DLBS | 32 | 0.490 | 2400 |
>     | DLBS-LA | 8 | **0.497** | 2500 |
>     | DLBS-LA | 16 | **0.517** | 4900 |
>
>     - Wan 2.1 1.3B, 81 frames (5s), MovieGen prompts
>     | Method | $KB$ | Reward | Inference Compute (NFE) |
>     |---|---|---|---|
>     | Vanilla | 1 | 0.313 | 50 |
>     | BoN | 128 | 0.357 | 6400 |
>     | BoN | 256 | 0.360 | 12800 |
>     | DLBS | 16 | 0.357 | 1200 |
>     | DLBS | 32 | 0.371 | 2400 |
>     | DLBS-LA | 8 | **0.373** | 2500 |
>     | DLBS-LA | 16 | **0.393** | 5000 |
>
> - Higher-resolution video generation
>
>     The resolution of the CogVideoX & Wan experiments conducted in **Section 5.3** is 832 * 480 > 512 * 512.
>     The experimental configurations are shown in **Appendix B**.
>
> **> Q3 & W4**
>
> > Does DLBS reduce diversity due to beam pruning toward high-reward paths?
>
> Please see **Figure 24** in **Appendix L.7**, where we have reported the reward–diversity trade‑off.
> DLBS achieves high performance while maintaining higher diversity than either BoN or GS.
> This exhibits a benefit from the wider possession and exploration of diffusion latents in DLBS, compared to BoN or GS.
>
> [1] Hunyuan Foundation Model Team. HunyuanVideo: A Systematic Framework For Large Video Generative Models. arXiv2025.

---

> ### Comment · Reviewer_3kkj · 2025-08-06
>
> Thank the authors for the rebuttal. The additional experiments and clarifications are appreciated and have addressed most of my concerns.
>
> However, I still have concerns about the sensitivity of performance to the reward weight configuration. While the authors show some transferability, the paper still lacks a clear strategy or guideline on how to select or adapt reward weights in practice.
>
> Considering the overall quality of the paper, I will maintain my positive score.

---

> ### Author Response · Authors · 2025-08-07
>
> We thank the reviewer for the constructive follow-up comment.
>
> **>Re: W5**
>
> > However, I still have concerns about the sensitivity of performance to the reward weight configuration. While the authors show some transferability, the paper still lacks a clear strategy or guideline on how to select or adapt reward weights in practice.
>
> For the guideline on how to select or adapt reward weights in practice, we would like to note that the procedure for reward calibration is clearly described in **Section 4.1**. To reflect the multi-dimensional nature of preferred videos, we model the calibrated reward function $r^*(\cdot, \cdot)$ as a weighted linear combination of various video quality metrics:
>
> $r^{*}(x_0, \mathbf{c}) := \sum_{i=1}^{M} w_i r_i(x_0, \mathbf{c})/\sum_{i=1}^{M} w_i$.
>
> The weights $w_i$ are determined by maximizing the Pearson correlation between the weighted reward and preference scores provided by VLMs (Gemini, GPT-4o). While the original paper adopts a brute-force search to determine these weights, we also experimented with learning $w_i$ as trainable parameters using Adam. The resulting reward correlations showed negligible differences from the brute-force approach (both approach shows R=0.51, p<0.01 for MSRVTT-test prompts).
>
>
> Lastly, we would like to share the concrete criteria based on our experimental observations; when a subset of prompts similar to the test prompt is available, it is sufficient to perform reward calibration on that subset to obtain appropriate weights (for in-domain generalizability, please see **W2**).
> If no such similar prompt sets are available, we suggest two practical alternatives: (1) use the universal reward derived during the rebuttal period, or (2) directly employ high-quality reward models such as VideoScore or VLMs.
>
> 1. **Universal Reward**
>
>     To clarify the universally generalized reward, we expanded the coverage of prompts used for calibration by combining four distinct prompt sets that vary in dynamics and difficulty: DEVIL-high, DEVIL-medium, DEVIL-static, and MSRVTT-test. This resulted in a unified set of 91 prompts used for calibration, yielding the following reward weights:
>
>     | Reward Metric | Subject Consistency | Motion Smoothness | Dynamic Degree | Aesthetic Quality | Text-Video Consistency |
>     |---|---|---|---|---|---|
>     | Weight | 0.5 | 0.5 | 0.5 | 0.5 | 0.75 |
>
>     We confirmed that using this Unified Reward in DLBS consistently improved VideoScore across diverse prompt sets:
>
>     | Method / VideoScore | DEVIL-high | DEVIL-medium | DEVIL-static | MSRVTT-test |
>     |---|---|---|---|---|
>     | Vanilla | 2.50 | 2.59 | 2.62 | 2.40 |
>     | DLBS ($KB=8$) + Unified Reward | 2.60 | 2.65 | 2.68 | 2.54 |
>
>     This demonstrates the feasibility of providing a reward configuration that generalizes well across prompt types.
>
> 2. **Use of VideoScore or VLMs Directly**
>
>     To demonstrate the applicability of VideoScore as a reward function for our DLBS, we conducted experiments directly using VideoScore as the reward function in DLBS during this rebuttal period. The following results imply that, in principle, high-performance VLMs available only through APIs (Gemini, GPT-4o) or actual humans can also be used as reward functions.
>
>     | Method | VideoScore |
>     |---|---|
>     | Vanilla | 2.40 |
>     | DLBS-LA ($KB=4$, $T'=6$) + VideoScore | 2.69 |
>
>
> As a supplement, we would like to note that reward weighting has a trend influenced by the dynamics specified in the prompt. As shown in **Figure 4** and **Appendix I.1**, the prioritization of video quality aspects varies according to the temporal characteristics of the prompt. To illustrate this, we analyzed the proportion of dynamics-related metrics (Dynamic Degree), temporal consistency (Subject Consistency, Motion Smoothness, Aesthetic Quality), and text-to-video alignment (Text-to-Video Consistency) in the calibrated reward weights derived from Gemini and GPT-4o:
>
> | Metric | VLM | DEVIL-high | DEVIL-medium | DEVIL-static | MSRVTT-test |
> |---|---|---|---|---|---|
> | Dynamics | Gemini | 0.167 | 0.125 | 0.000 | 0.167 |
> |  | GPT-4o | 0.333 | 0.250 | 0.250 | 0.273 |
> | Temporal Consistency | Gemini | 0.167 | 0.750 | 0.400 | 0.583 |
> |  | GPT-4o | 0.000 | 0.500 | 0.250 | 0.364 |
> | T2V Alignment | Gemini | 0.667 | 0.125 | 0.600 | 0.250 |
> |  | GPT-4o | 0.667 | 0.250 | 0.500 | 0.364 |
>
> These results confirm that the calibrated rewards place greater emphasis on Dynamics for prompts with high motion (e.g., DEVIL-high). At the same time, Temporal Consistency becomes more important for static or less dynamic prompts. Regardless of the prompt characteristics, however, Text-to-Video Alignment remains a consistently emphasized component across all settings.
>
> To make it clear for the reader, we will include these discussions in the revised paper.

---

### Official Review · Reviewer_srve · 2025-07-02

**Clarity:** 3
**Significance:** 2
**Originality:** 3
**Rating:** 4
**Confidence:** 4

**Summary:**

This paper introduces latent beam search at inference time to align the generated videos with input prompts. There are two main contributions: 1) It proposes latent beam search and lookahead estimator for inference-time scaling. It samples K latents and selecting B beams by evaluating the reward through lookahead estimator. 2) It proposes a reward calibration algorithm to better evaluate the quality of generated videos.

**Questions:**

1) Computational inefficiency and limited improvement. According to Figure.1 (right) and Figure.6 (left), with $2^5$x increase in inference computation time, the improvement is only around 10%. Given the additional computation required, I think the improvement is far from sufficient. Moreover, although the proposed method shows improved performance with increased inference time, the simplest BoN baseline also exhibits a similar trend, and the improvement over BoN is also limited (less than 10%). Therefore, I believe the proposed method gives very limited improvement in test-time scaling performance.
2) Beam search is still random search. At each timestep, the method still introduces random noise to generate different beam candidates. This is essentially not different from Best of N (BoN), except that it expands the space of random search. BoN's randomness lies in the initial latent, while beam search introduces randomness at every timestep. I believe the effectiveness of such random search is inherently limited. Although the authors use classifier guidance to show that the search process is moving toward the target, the zero-order nature of the random search seems not enough. If it were possible to compute first- or higher-order gradients with respect to the target and update accordingly, I believe that would be more meaningful.

**Ethical Concerns:**

["NO or VERY MINOR ethics concerns only"]

**Final Justification:**

The authors adrress my concern in computational inefficiency and the difference between BoN, zero-order search and first-order search.

Overall, I believe that the test-time scaling of diffusion models is a very interesting and important academic problem. While there is still much to be addressed, I think this paper takes at least a small step in the right direction. I will raise my score and lean towards accepting it. Thank you once again for the authors' continued responses.

**Limitations:**

Yes

**Quality:**

2

**Strengths And Weaknesses:**

Strength:
1) The problem explored in this paper: how to perform test-time scaling in diffusion models is a very meaningful one. The proposed beam search method is straightforward and solid.
2) This paper's writing is good and the method is easy to understand.

Weaknesses:
See questions.

---

> ### Author Rebuttal · Authors · 2025-07-31
>
> We appreciate the reviewers’ insights and address each point below.
>
> **> Q1**
>
> > I believe the proposed method gives very limited improvement in test-time scaling performance.
>
> While 10% reward increase is seemingly small, it represents a strong perceptual quality improvement. Here, we conduct human evaluations comparing DLBS-LA (KB=8, T’=6, NFE=2500) and BoN (KB=64, NFE=3200).
> DLBS-LA significantly outperforms BoN in human evaluation despite requiring fewer NFEs.
>
> | Model | Prompt Set | Win Rate of DLBS-LA against BoN | Reward (DLBS-LA) | Reward (BoN) |
> |---|---|---|---|---|
> | Latte | DEVIL-medium | **69.70%** | 0.726 | 0.705 |
> | Latte | MSRVTT-test | **65.56%** | 0.607 | 0.578 |
> | CogVideoX-5B | DEVIL-very-high | **69.70%** | 0.500 | 0.476 |
> | Wan 2.1 14B | MovieGen | **60.00%** | 0.378 | 0.357 |
>
> Moreover, in the original paper, the approximately 10% reward improvement achieved by DLBS has been confirmed to enhance human preference and human preference-trained evaluation models (VideoScore) across multiple prompt sets and models (**Section 5.2**). We also have confirmed that DLBS is superior to  Best-of-N (BoN) when using VideoScore across multiple datasets (**Figure 7; Middle**).
>
> **> Q2**
>
> > I believe the effectiveness of such random search is inherently limited. Although the authors use classifier guidance to show that the search process is moving toward the target, the zero-order nature of the random search seems not enough.
>
> One clear advantage over gradient-based methods is the ability to use reward models where gradients are either computationally prohibitive (e.g., large-scale VLMs that are not scalable for backpropagation) or fundamentally impossible to compute (e.g., human evaluators or external API-based models).
> To simulate the search with human feedback, we conducted a search using VideoScore, a VLM-based reward model trained on human evaluations, and evaluated the results with human feedback.
> The following results imply that, in principle, high-performance VLMs available only through APIs (Gemini, GPT-4o) or actual humans can also be used as reward functions.
>
> | Method | VideoScore |
> |---|---|
> | Vanilla | 2.40 |
> | DLBS-LA ($KB=4$, $T’=6$) + VideoScore | **2.69** |
>
> Furthermore, to address reviewer concerns about the efficiency of our proposed method when using gradient-computable reward models, we compare our DLBS against  DAS [1], a first-order gradient-based search method based on Sequential Monte Carlo, where N represents the number of particles, and compare the metric improvements relative to runtime.
> Runtime measurements were conducted on NVIDIA RTX 6000 Ada (48GB).
> Due to computational cost constraints as described later, we compared using SD 1.5 with LAION Aesthetic V2 as the reward model.
>
> | Method | Score | Time (s) | Memory (GB) |
> |---|---|---|---|
> | Vanilla | 5.81 | 2 | 4.3 |
> | DAS N=8 | 6.59 | 108 | 14.2 |
> | DLBS K=8, B=2 | **6.63** | **103** | **5.9** |
>
> DLBS achieves competitive results compared to first-order gradient-based search methods within similar computational time.
>
> In addition, gradient-based search methods exhibit significant memory usage increases due to computations for the reward function and VAE decoder. In other words, gradient-based methods are actually inefficient in terms of memory cost.
> Furthermore, these results are for a single evaluator with a single frame (i.e., image generation).
> Note that for video generation, as mentioned in **Figure 3**, a single evaluator metric does not correlate well with perceptual quality, necessitating the combination of multiple evaluators, which roughly multiplies the memories for back propagation $N_{\mathrm{reward}}$ times.
> Additionally, video generation models do not decode just one frame from the VAE (maximum frames are 81 for Wan 2.1 and 49 for CogVideoX).
> The gradient increase would be significantly larger in video generation than in image generation, making gradient-based methods almost impossible in practice.
>
> For reference, the memory usage of vanilla Latte, DLBS-LA, and DAS is as follows:
> | Method | Vanilla | DLBS-LA | DAS |
> |---|---|---|---|
> | Memory Usage (GB) | 16.3 | 38.9 | > 48.0 (OOM) |
>
> [1] Kim et al. Test-time Alignment of Diffusion Models without Reward Over-optimization. ICLR2025.

---

> > ### Comment · Reviewer_srve · 2025-08-05
> >
> > I appreciate the auther's additional experiments and detailed explanation.
> >
> > For the first question, I understand that DLBS-LA outperforms BoN both in human evaluation and reward. My main concern lies in the substantial additional computational resources required to achieve quality improvements compared to the vanilla method, raising the question of whether such trade-offs are justifiable.
> >
> > For the second question, I understand that the performance between our method and the first-order method are competitive, and our method is more efficient (less time and VRAM required). Are there reasonable explanations for why first-order and zero-order methods exhibit similar performance? In my understanding first order should outperform zero-order method.

---

> ### Author Response · Authors · 2025-08-06
> **Official Comment by Authors (1/2)**
>
> We thank the reviewer for the follow-up comment.
>
> **> Re: Q1**
>
> > My main concern lies in the substantial additional computational resources required to achieve quality improvements compared to the vanilla method, raising the question of whether such trade-offs are justifiable.
>
> As discussed in the limitations (**Section 7**), our paper proposes a method that improves generation quality by investing computational resources at inference time. We believe that such investment is acceptable and worth considering for the following reasons:
>
> - In some applications, generation quality is prioritized over inference speed. Particularly when generating short clips for advertising purposes, it is a natural requirement to prioritize visual appeal, even if it requires more generation time. While this may not apply to all applications, we believe the demand is substantial enough to justify this line of research.
>
> - When considering actual video generation use cases, it is common to perform explicit Best-of-N (BoN) by generating with multiple seeds and selecting the best-looking one among them, or perform implicit sequential BoN by repeatedly generating until we obtain satisfactory results (so-called "cherry-picking" or "Gacha-style" generation). Since our DLBS-LA achieves superior performance with less computational cost than BoN (please see **Figure 6 (Left)**, **Q1**), it can efficiently replace the inference-time search methods that are customarily accepted in this field. This point has been discussed in the limitations section (**Section 7**).
>
> - In the field of image generation, it is known that investing computational cost at inference time is more efficient than during post-training [2]. In the rebuttal period, we confirmed that a similar phenomenon can be observed in video generation as well. As shown in the table below, applying one of the stable post-training methods, VideoDPO [3], to VideoCrafter2 yields negligible performance improvements. However, when combined with our DLBS, the performance improves significantly across both datasets, DEVIL-high and MSRVTT-test.
>
>     | Method/Prompt | DEVIL-high | MSRVTT-test |
>     |---|---|---|
>     | VideoCrafter2 | 0.337 | 0.555 |
>     | + DPO | 0.335 | 0.556 |
>     | + DPO & DLBS | **0.359** | **0.576** |
>
> Of course, we consider the development of more computationally efficient search methods, as well as integration with orthogonal research aimed at accelerating inference, to be promising directions for future work.
>
> [2] Liu, et al. VideoDPO: Omni-Preference Alignment for Video Diffusion Generation. CVPR2025.
> [3] Ma, et al. Inference-Time Scaling for Diffusion Models beyond Scaling Denoising Steps. CVPR2025.

---

> ### Author Response · Authors · 2025-08-06
> **Official Comment by Authors (2/2)**
>
> **> Re: Q2**
>
> >  Are there reasonable explanations for why first-order and zero-order methods exhibit similar performance?
>
> First, our observation that DLBS (a zero-order method) and DAS (a first-order method) achieve comparable performance aligns with findings from prior work on inference-time search for image generation [4].
> In [4], first-order gradient-based guidance and a zero-order Sequential Monte Carlo-based method demonstrated equivalent performance, indicating that this is not a novel trend we are reporting for the first time.
> One possible explanation is that computing first-order gradients requires mapping from the latent space to the image space using Tweedie’s formula.
> Since reward value estimation is extremely noisy, the gradient noise also becomes large, which may prevent efficient maximization of rewards during evaluation.
>
> Second, during the rebuttal period, we conducted additional experiments for image generation under increased computational cost.
> Specifically, due to the higher computational efficiency of DLBS, it can afford approximately twice the sampling budget compared to DAS when execution time is equalized.
> Note that sampling budget means $K*B$ for DLBS, and $N$ for DAS.
> Here, when twice the execution time was allowed, i.e., comparing DLBS $K=8$, $B=2$ with DAS $N=16$, we observed a trend where DAS outperformed. However, when DLBS execution time was kept the same, with DLBS $K=16$, $B=2$, this advantage disappeared.
>
> | Method | Score | Time (s) | Memory (GB) |
> |---|---|---|---|
> | Vanilla | 5.81 | 2 | 4.3 |
> ||||
> | DAS $N=8$ | 6.59 | 108 | 14.2 |
> | DLBS $K=8$, $B=2$ | **6.63** | **103** | **5.9** |
> ||||
> | DAS $N=16$ | 6.68 | 220 | 14.2 |
> | DLBS $K=16$, $B=2$ | **6.69** | **209** | **5.9**  |
>
> In other words, we confirmed that under the assumption of equal sampling budget, the advantage of first-order gradients is observed. In contrast, under the assumption of equal execution time, zero-order gradients take the lead because they can have a larger search budget.
>
> Finally, we emphasize that all additional experiments comparing the first-order and zero-order methods were conducted in the context of image generation, whereas the primary focus of this paper is video generation.
> In the extended experiments presented during the rebuttal (**Q2**), we demonstrated that applying first-order gradient methods to video generation poses significant practical challenges—such as out-of-memory errors even on 48GB GPUs—and that no feasible algorithms have yet been established for this setting.
> Moreover, extrapolating from the image generation results, we argue that even if such feasibility issues were addressed, our zero-order method can still achieve substantial quality improvements and has the potential to perform comparably to, or even surpass, first-order methods.
>
> Discussion on the above points will be included in the revised paper.
>
> [4] Singhal, et al. A General Framework for Inference-time Scaling and Steering of Diffusion Models. ICML2025.

---

> > ### Comment · Reviewer_srve · 2025-08-06
> >
> > I really appreciate the author's reply. I believe I now understand and acknowledge the proposed method’s application, use cases, and its advantages over the baselines.
> >
> > I would like to ask one final question: In the first paragraph of Re: Q2, you mentioned another phenomenon where zero-order and first-order methods perform similarly. Your explanation was that “since reward value estimation is extremely noisy, the gradient noise also becomes large, which may prevent efficient maximization of rewards during evaluation.” Can I simply understand it as: current reward models are not good enough and the provided rewards are not accurate and stable.

---

> > > ### Author Response · Authors · 2025-08-07
> > >
> > > We really thank the reviewer for the additional follow-up comment.
> > >
> > > > **Re: Re: Q2**
> > > > Can I simply understand it as: current reward models are not good enough and the provided rewards are not accurate and stable?
> > >
> > > Our point is not that current reward models are fundamentally inadequate or that the rewards are inherently inaccurate or unstable. Rather, the key challenge in inference-time search (for both zeroth-order and first-order) lies in the evaluation of noisy intermediate latents during the diffusion process, as already discussed in our original paper (**Section 3.1**).
> > >
> > > The search methods discussed so far — including both first-order and zero-order approaches — operate in the latent space of latent diffusion models, where the original image $x$ is first encoded into a latent representation $z_0 = \text{Enc}(x) $, and the diffusion model learns to denoise from a Gaussian noise $z_T$ to $z_0$.
> > > The search methods perform a search over intermediate latents $z_t$ to guide generation.
> > >
> > > However, $z_t$ is not a fully denoised latent, and decoding it directly yields degraded outputs. Most reward functions $r(x)$ are designed to assess clean, fully denoised images and are not robust to such noisy inputs.
> > >
> > > To address this, most of the prior works [1,5,6] often use Tweedie's formula to approximate the denoised latent $ \hat{z_{0|t}} $ from $z_t$ via a single-step projection:
> > >
> > > $\hat{z_{0|t}} = \frac{1}{\sqrt{\alpha_t}} \left( z_t - \sqrt{1 - \alpha_t} \epsilon_\theta(z_t, t) \right)$,
> > >
> > > where $\epsilon_\theta(z_t, t)$ is the predicted noise and $\alpha_t$ is a noise scheduling parameter. Since this formula approximates the result of the remaining $t$ denoising steps, it introduces residual noise due to the single-step approximation. As a result, the decoded image $ \text{Dec}(\hat{z_{0|t}}\) $ remains coarse, and the reward evaluation $ r(\text{Dec}(\hat{z_{0|t}}\)\) $ may be noisy and unstable.
> > >
> > > Consequently, first-order methods that rely on gradients from such approximated rewards may suffer from inaccurate reward maximization as their optimization signal is corrupted by projection noise.
> > >
> > > Motivated by this observation, we have proposed the Lookahead Estimator in the original paper, which performs multi-step denoising before reward evaluation, allowing for more accurate and stable feedback for optimization (**Section 3.2**).
> > >
> > > [5] Yu, et al. FreeDoM: Training-Free Energy-Guided Conditional Diffusion Model. ICCV2023.
> > > [6] Bansal, et al. Universal Guidance for Diffusion Models. ICLR2024.

---

> > > > ### Comment · Reviewer_srve · 2025-08-08
> > > >
> > > > Thank you for your response. I understand why first-order and zero-order methods have similiar performance.
> > > >
> > > > Overall, I believe that the test-time scaling of diffusion models is a very interesting and important academic problem. While there is still much to be addressed, I think this paper takes at least a small step in the right direction. I will raise my score and lean towards accepting it. Thank you once again for the authors' continued responses.

---

### Official Review · Reviewer_rNpU · 2025-07-03

**Clarity:** 3
**Significance:** 3
**Originality:** 3
**Rating:** 4
**Confidence:** 3

**Summary:**

The paper tackles the problem of alignment in text-to-video diffusion, making generated videos both follow a text prompt and high perceptual quality in inference time.
The Diffusion Latent Beam Search and a lookahead estimator are proposed to obtain a suitable sample at each denoising step.
The authors also introduced a weight linear combination of multiple metrics that aligned with perceptual quality and human preference.

**Questions:**

1. The LA estimator selects $T^\prime=6$. Does it work for different models? Does it hold for different samplers?
2. Beam size and candidate count interact with complexity. Have you explored any dynamic or adaptive strategies to adjust ($B, K$) on-the-fly based on intermediate reward?

**Ethical Concerns:**

["NO or VERY MINOR ethics concerns only"]

**Final Justification:**

This paper is the first to investigate inference time beam search for diffusion video generation, presenting extensive experiments that demonstrate its superior quality compared to naive generation. It also explores strategies for performing beam search efficiently. I recommend accepting this paper.

**Limitations:**

The beam search can reduce the inference cost compared to best-fo-N. But the still takes tens of seconds per video on high-end GPUs, limiting real-time or large-scale use.

**Paper Formatting Concerns:**

I have no concerns regarding the formatting.

**Quality:**

3

**Strengths And Weaknesses:**

strength:
1. the paper proposed a beam search for inference-time alignment in the latent space.
2.  systematic study of six perceptual metrics correlations with VLM/Human preferences.
3. extensive experiments on different parameters (e.g., $B, H, T^\prime$)
4. Effectively improve the performance across different diffusion model variants.

---
weakness:
1. calibration relies on a strong VLM (e.g., Gemini, GPT). Without these models, it might be challenging to construct reliable metrics.
2. efforts of tuning ($B, K, T^\prime$) can be expensive.

---

> ### Author Rebuttal · Authors · 2025-07-31
>
> We appreciate the reviewer’s thoughtful feedback.
>
> **> W1**
> > calibration relies on a strong VLM (e.g., Gemini, GPT). Without these models, it might be challenging to construct reliable metrics.
>
> As explained in **Section 4**, while our current approach uses VLM feedback for reward calibration, this is primarily for practical demands rather than a fundamental necessity.
>
> Human evaluation remains the gold standard for video quality assessment, but collecting human feedback at the required scale for search is prohibitively expensive and time-consuming.
>
> A practical approach to reduce time and cost is leveraging AI feedback from VLMs, which has shown modest alignment with human judgment on video quality (**Appendix J**).
> Since alignment via inference-time search requires massive reward evaluation queries, we need tractable proxy rewards that avoid reliance on humans or external VLM APIs for deployment.
> Existing lightweight metrics for video naturalness (e.g., Subject Consistency, Dynamic Degree) often fail to correlate with human/VLM evaluations (**Figure 3**), motivating our reward calibration approach that weights existing metrics using VLMs.
> While human labels would be ideal for calibration, collecting the extensive metric evaluations required would be even more costly and time-consuming than gathering preference labels from VLM APIs.
> || Human | VLM | Lightweight Metrics |
> |---|---|---|---|
> | Alignment with Human Eval. | ◎ | ○ | △ |
> | Cost | △ | ○ | ◎ |
> | Exec. Speed | △ | ○ | ◎ |
>
>
> Crucially, to demonstrate the feasibility of reward calibration without VLM APIs, we performed calibration on MSRVTT-test prompts using VideoScore (a reward model trained on human evaluations) and searched accordingly, showing that our approach can work in principle without external VLM dependencies.
>
> | Method | VideoScore |
> |---|---|
> | Vanilla | 2.40 |
> | DLBS-LA ($KB=4$, $T’=6$) + VideoScore Calibrated Reward | **2.60** |
>
> **> W2**
>
> > efforts of tuning (B,K,T′) can be expensive.
>
> - ($K$, $B$)
>
>     In **Appendix L.3** and **L.9**, we present ablation studies on (B,K) allocation across multiple prompts and models.
>     As described in **Appendix L.3**, we found that the optimal allocation typically peaks around 25%-50%.
>     We also demonstrate that this pattern holds not only for video generation but also for image generation (**Appendix L.12**).
>     These findings provide practical guidance, suggesting that exploration can be focused within this specific range rather than exhaustively searching the entire parameter space.
>
> - $T'$
>
>     We comprehensively validated across multiple prompt sets, models, and samplers that adopting $T' \approx 6$ achieves performance improvements, providing clear guidelines (please see **Appendix L.4** and **Q1**).
>
> **> Q1**
>
> > The LA estimator selects $T′=6$. Does it work for different models? Does it hold for different samplers?
>
> To further verify that the choice of $T'=6$ is effective beyond DDIM solvers, we conducted an ablation study with SDE-DPMSolver++ for Wan 2.1 1.3B, varying $T'$ across $1, 2, 3, 6$, and $12$. Even in this case, performance improved efficiently up to $T'=6$, achieving nearly identical performance to $T'=12$, which requires twice the computational cost.
> This supports that the choice of $T'=6$ works for different solvers (As we show in DDIM in **Appendix L.4** of the paper).
>
> | $T'$ | 1 | 2 | 3 | 6 | 12 |
> |---|---|---|---|---|---|
> | Reward | 0.351 | 0.360 | 0.365 | **0.376** | **0.380** |
>
> Moreover, as shown in **Figure 6**, we confirmed that `DLBS-LA` with $T'=6$ achieves superior performance compared to methods without LA estimator (i.e., `DLBS`, `GS`, `BoN`) across different models (Wan 2.1, CogVideoX, Latte).
>
> **> Q2**
>
> > Have you explored any dynamic or adaptive strategies to adjust ($B$,$K$) on-the-fly based on intermediate reward?
>
> We initially considered dynamic or adaptive strategies that tune the allocation of ($B$,$K$) across diffusion timesteps, such as starting with a larger $B$ in early steps and transitioning to emphasize $K$ as the process progresses. However, we did not prioritize this direction because of the following reasons; fIrst, the performance gains from incorporating the lookahead estimator (please compare `DLBS` and `DLBS-LA` in **Figure 5; Left**) are significantly larger than those from tuning ($B$,$K$) allocation (**Figure 5; Right**) (please see **Appendix L.3, L.9** for further results).
> We also compared against Sequential Monte Carlo-based methods that adaptively decide whether to search or retain previous candidates at each diffusion step, but these showed weaker performance improvements than DLBS (**Appendix L.8**). Adaptive meta-planning on how to allocate $KB$ budget would be an interesting future direction.

---

> > ### Comment · Reviewer_rNpU · 2025-08-05
> >
> > Thanks for the author's reply. I appreciate additional comparison and detailed explanation. I will remain the score.

---

### Official Review · Reviewer_3Psp · 2025-07-04

**Clarity:** 3
**Significance:** 3
**Originality:** 3
**Rating:** 4
**Confidence:** 4

**Summary:**

The paper addresses a challenge of perceptual misalignment in text-to-video diffusion models, where generated videos often contain unnatural motion, temporal inconsistencies or static scenes. The authors propose Diffusion Latent Beam Search (DLBS) with lookahead estimator, an inference-time method that optimizes video quality by selecting a better diffusion latent to maximize a given alignment reward. DLBS leverages beam search over diffusion latents guided by a lookahead estimator to reduce noise in reward evaluation. The authors also provide a technique for reward calibration by leveraging AI feedback from VLMs. The higher a strong VLM estimates the generated video quality, the better visual quality it would have from a human point of view. Six base reward functions for perceptual video quality are used: Subject Consistency, Motion Smoothness, Dynamic Degree, Aesthetic Quality, Imaging Quality and Text-Video Consistency. At the experimental stage the authors demonstrate that DLBS is the most scalable, efficient and robust inference-time search that significantly improves video quality under the same computational costs.

**Questions:**

I would advice to make additional comments on generalization of VLM-Human correlation and reward calibration to improve the paper strengths. Ablations on solvers (e.g., DPMSolver++) or nu values would clarify optimization choices instead of just using deterministic DDIM (nu=0) at the experimental setup.

**Ethical Concerns:**

["NO or VERY MINOR ethics concerns only"]

**Final Justification:**

I will remain my score according to the authors’ answers

**Limitations:**

Yes

**Paper Formatting Concerns:**

No paper formatting concerns detected

**Quality:**

3

**Strengths And Weaknesses:**

Strengths.

The authors provide a strong vision to better denoising way in latent space which is the core contribution of the paper. The ablations and correlations are thorough. Human evaluations (pairwise comparisons) and diverse prompts (DEVIL, MSRVTT, MovieGen) strengthen validity.


Weaknesses.

While VLMs are used as human proxies, their correlation with human evaluation is moderate (nearly 0.5 correlation). Relying on VLMs without deeper analysis of failure cases (e.g., prompt ambiguity) provides additional risks for reward over optimization and highly biasing. The computational cost is not presented at the experimental stage. The rewards use fixed weights per prompt category after calibration, but prompt adaptation is unexplored.

Clarity. The text is well-structured and mostly clear. Figures effectively illustrate the method and results. The pseudocode and appendices sections aid reproducibility.

Significance. The paper addresses a critical gap in text-to-video generation. The inference time approach is model-agnostic, scalable (tested up to 14B params) and deployable without additional training or fine-tuning. The efficiency claims show readiness to real-world use.

Originality. DLBS adapts a well-known beam search paradigm to diffusion latents, while reward calibration and lookahead estimation offer novel solutions to reward noise and misalignment.

---

> ### Author Rebuttal · Authors · 2025-07-31
>
> We thank the reviewer for the constructive feedback.
>
> **> W1**
>
> > Relying on VLMs without deeper analysis of failure cases (e.g., prompt ambiguity) provides additional risks for reward over optimization and highly biasing.
>
> First, for a deeper analysis of failure cases, we qualitatively examined the top 5% outliers between human preference labels used in **Appendix J** and VLM (Gemini) linear regression predictions.
> - For prompts specifying text, such as "the words 'KEEP OFF THE GRASS' on a sign next to a lawn," the VLM was significantly harsher than human evaluation on text rendering (VLM: 2/10, Human: 4.5/5).
> - For prompts specifying quantities, such as "Four friends have a picnic, enjoying six sandwiches and two bottles of juice," the VLM was more lenient than human evaluation (VLM: 8/10, Human: 1.5/5).
> - For "fashion portrait shoot of a girl in colorful glasses, a breeze moves her hair," despite missing arms in the generated person, the VLM was misled by distracting background patterns, possibly mistaking them for curtain-like elements that obscure the arms behind the background (VLM: 8/10, Human: 1.2/5).
>
> As far as we observed, VLM sometimes makes subtle mistakes, but we did not see any critical failures.
>
> Next, to avoid the biases from specific VLMs, we have analyzed preference trends for GPT-4o and Gemini. In **Appendix I.1**, we present reward calibration weights using GPT-4o and compare them with Gemini calibration results (**Figure 4**), analyzing that GPT-4o tends to emphasize dynamics more than Gemini in the trade-off between dynamics and consistency.
> To make this clearer, we calculated the proportion of dynamics metrics (i.e., Dynamic Degree), temporal consistency (i.e., Subject Consistency, Motion Smoothness, and Aesthetic Score), T2V alignment (i.e., Text-to-Video Consistency) in the calibrated reward weights using Gemini and GPT-4o.
> | Metric | VLM | DEVIL-high | DEVIL-medium | DEVIL-static | MSRVTT-test |
> |---|---|---|---|---|---|
> | Dynamics | Gemini | 0.167 | 0.125 | 0.000 | 0.167 |
> |  | GPT-4o | 0.333 | 0.250 | 0.250 | 0.273 |
> | Temporal Consistency | Gemini | 0.167 | 0.750 | 0.400 | 0.583 |
> |  | GPT-4o | 0.000 | 0.500 | 0.250 | 0.364 |
> | T2V Alignment | Gemini | 0.667 | 0.125 | 0.600 | 0.250 |
> |  | GPT-4o | 0.667 | 0.250 | 0.500 | 0.364 |
>
> While GPT-4o emphasizes dynamics more and Gemini prioritizes temporal consistency, both show similar text alignment weighting.
>
> Lastly, regarding concerns about reward overoptimization, we confirmed in the **Appendix L.7** that diversity is not reduced by reward optimization, and we have verified that our method improves performance on both automatic evaluations, different from calibrated reward to be optimized, and human evaluations (**Section 5.2**).
>
> **> W2**
>
> >  The computational cost is not presented at the experimental stage.
>
> - For search algorithms
>
>     We provide the types and numbers of GPUs used in the **Appendix B.4**.
>     For Latte, we present computational costs (please see **Figure 6; Left**).
>     For CogVideoX and Wan 2.1, we newly demonstrate the relationship between cost and performance when searching with maximum frame counts (please see **Q2 from Reviewer 3kkj**).
>
> - For reward calibration
>
>     As mentioned in **Section 4.1**, we generate 64 videos per prompt from pre-trained Latte and Wan 2.1.
>     Our reward calibration approach is significantly more cost-efficient than naively querying VLMs every time during search. This is demonstrated in the following comparison between the per-prompt query count of reward calibration and that of naive VLM-based reward evaluation under DLBS with Wan 2.1 1.3B ($KB=32$).
>
>     | Method | Query Count | Exec. Time (sec) |
>     |---|---|---|
>     | Reward Calibration | $64$ | $\approx$ 960 |
>     | Querying VLMs during Search ($KB=32$) | $(T=)50 \times (KB=)32 = 1600$ | $\approx$ 102,400 |
>
> **> W3**
>
> > The rewards use fixed weights per prompt category after calibration, but prompt adaptation is unexplored.
>
> Our experiments have shown that optimizing an existing single reward is not aligned with video preferences (**Figure 3**), and that we should consider the weighted combination of existing rewards to be a better proxy for judging preferable videos. Because the important metrics for a preferred video can differ among the prompts (e.g., focusing on dynamic movement, aesthetic static scene, etc), we have suggested that reward calibration should be done per prompt categories, where they have similar dynamics. The individual prompt is the smallest unit of the prompt categories, so our experiments support that it is possible to adapt rewards into specific prompts.
>
> Moreover, although we fixed the weight of rewards per category in the paper,  we here demonstrate that expanding prompt coverage enables more flexible reward design.
> While the paper shows reward calibration performed separately on each prompt set (DEVIL-high, medium, static, and MSRVTT-test), we now conduct a unified calibration approach.
> We combine all four prompt sets into a single dataset of 91 prompts and perform reward calibration on this comprehensive collection.
> The resulting calibrated reward is then applied to evaluate performance on each individual prompt set, demonstrating improved generalizability across diverse prompt types.
>
> | Method/VideoScore | DEVIL-high | DEVIL-medium | DEVIL-static | MSRVTT-test |
> |---|---|---|---|---|
> | Vanilla | 2.50 | 2.59 | 2.62 | 2.40 |
> | DLBS ($KB=8$) + Unified Reward | 2.60 | 2.65 | 2.68 | 2.54 |
>
> In summary, we believe that our reward calibration can achieve either specialization and generalization, depending on the given prompts and how we want to optimize the reward.
>
> **> Q1**
>
> > I would advice to make additional comments on generalization of VLM-Human correlation and reward calibration to improve the paper strengths.
>
> - Generalization of VLM-Human correlation
>
>     The validity of using VLM evaluation as a proxy for human evaluation is not solely supported by the moderate correlation between evaluations.
>     Most importantly, **Section 5.2** demonstrates that search using VLM-calibrated rewards successfully improves Human Preference and Human Preference-trained Reward Models (VideoScore) across multiple prompt sets and models.
>
>     To further strengthen this argument, we conducted human evaluations comparing DLBS-LA ($KB=8$, $T’=6$, $\mathrm{NFE}=2500$) and BoN ($KB=64$, $\mathrm{NFE}=3200$) in **W3 from Reviewer 3kkj**. Because the calibrated rewards can work as proxy functions for human evaluation, DLBS-LA outperforms BoN in human evaluation despite requiring fewer NFEs.
>
>     These results support the effectiveness of using VLM evaluation as a substitute for human evaluation in improving human evaluation outcomes.
>
>     For discussion of trend in Gemini and GPT-4o, please refer to **W1**.
>
> - Generalization of reward calibration
>     - Generalization for prompts
>
>         We first note that video generation inherently involves trade-offs between fundamental properties such as dynamics and consistency (**Appendix G**), which may require category-specific calibration for optimal performance.
>         However, despite these domain-specific requirements, we hypothesize that calibrated rewards can generalize to some extent across different datasets, as they are based on shared principles of perceptual quality.
>          To test the out-of-domain transferability, we conducted additional experiments applying the calibrated reward weights calibrated on DEVIL-high and DEVIL-medium to MSRVTT-test prompts.
>         We evaluated the results using VideoScore, a human preference-trained evaluator, measuring five key metrics along with their average scores.
>
>         | (R: Reward, P: Prompt)| Visual Quality | Temporal Consistency | Dynamic Degree | T2V Alignment | Factual Consistency | Average |
>         |---|---|---|---|---|---|---|
>         | **Vanilla** | 2.32 | 2.01 | **2.91** | 2.67 | 2.07 | 2.40 |
>         | **DLBS (KB=8)** | | | | | | |
>         | + R=MSRVTT, P=MSRVTT | **2.50** | **2.27** | 2.88 | **2.74** | 2.28 | 2.53 |
>         | + R=DEVIL-med, P=MSRVTT | 2.49 | 2.26 | 2.89 | 2.73 | **2.31** | **2.54** |
>         | + R=DEVIL-high, P=MSRVTT | 2.36 | 2.03 | ***2.90*** | 2.68 | 2.10 | 2.42 |
>
>         With the DEVIL-high reward, we can enhance other metrics while maintaining dynamics.
>         DEVIL-medium reward, which is a closer domain to MSRVTT-test, shows a different trade-off pattern: while it slightly reduces dynamics, it significantly improves other metrics and achieves a higher average score than the MSRVTT-test reward, demonstrating higher transferability.
>
>         Moreover, in the original paper, we have confirmed that generalization for unseen in-domain prompts, using the same reward, can improve performance on unseen prompts (please see **Figure 8; Right**).
>
>     - Generalization for video length
>
>         Our method is scalable even when extending the frame length to the model's maximum.
>         To demonstrate this, we conducted additional experiments with the maximum frames for Wan 2.1 1.3B and CogVideoX-5B (please see **Q2 from Reviewer 3kkj**).
>         The reward, which was calibrated using 33 frames, 2s videos generated by Wan 2.1 1.3B, was applied as-is.
>         As a result, we confirmed that even with longer frames, the reward values could be improved more efficiently than the BoN baseline.
>
>     - For generalization for models and solver types, please see **Q2**.
>
> **> Q2**
>
> > Ablations on solvers (e.g., DPMSolver++) or nu values would clarify optimization choices instead of just using deterministic DDIM ($nu=0$) at the experimental setup.
>
> While used DDIM with $\eta=0$ for Latte calibration, DPMSolver++ was employed for Wan 2.1 calibration (please see **Appendix B.3** for detailed settings).
> Notably, we confirmed that rewards collected from Wan 2.1 1.3B with DPMSolver++ can improve human evaluation scores for CogVideoX 5B with DDIM solver (**Section 5.3, Appendix L.9**).

---

> > ### Comment · Reviewer_3Psp · 2025-08-05
> >
> > Thanks for the author's reply. I appreciate comparison additional experiments and results provided in the answer. I will remain the score

---

### Note · Authors · 2025-08-12

Dear Reviewers, AC, and SAC,

We thank you for your great effort in reviewing our paper.
Again, we highlight key points as follows. It would be great if you considered these for your final decision.

**Computational Efficiency** (**srve**)
Additional experiments confirm the computational efficiency of DLBS.
- Human evaluation shows that DLBS-LA outperforms Best-of-N with fewer NFEs.
- Compared to first-order methods (i.e., DAS), DLBS performs better under equal time and avoids the large memory overhead that makes first-order guidance impractical for video generation.
- Applying the post-training method VideoDPO to VideoCrafter2 yields negligible gains, whereas combining it with DLBS produces substantial improvements, showing inference-time search can outperform post-training.

These results show DLBS offers a superior quality–cost trade-off and is a scalable, efficient option for high-quality video generation.

**Clear Guidance of Reward Selection** (**3kkj**, **3Psp**)
- The reward calibration procedure is detailed in **Section 4.1**.
- Practical guidance: (1) when prompts similar to the test prompt are available, we calibrate on that subset; otherwise, (2) use the universal reward derived during the rebuttal period that consistently improve video quality across prompt sets, or (3) directly use strong evaluators (VideoScore/VLMs). Additional experiments support these strategies.
- Supplementary analysis: dynamic prompts prioritize dynamics metrics, static prompts prioritize temporal consistency, and text–video alignment is essential in all cases.

**Wide Range of Additional Experiments**
We have added various additional experiments to support our claims.
- Analyze failure cases of VLM evaluation via outliers between human scores and Gemini scores, finding only subtle errors (**3Psp**).
- Assess reward transferability: apply weights from DEVIL-high/medium to MSRVTT-test, consistently improving scores (**3kkj**, **3Psp**).
- Show calibration is possible without VLM APIs using VideoScore (**rNpU**, **3kkj**).
- Verify LA estimator's $T'$ choice generalizes beyond DDIM with SDE-DPMSolver++ (**rNpU**).
- Confirm scalability to longer videos with maximum length of Wan 2.1 and CogVideoX; DLBS still outperforms BoN with fewer NFEs (**3kkj**).

We have received replies from all reviewers and believe that we have addressed their concerns. Should there be any additional questions or concerns, we will be happy to respond.

Best regards,
The authors

---

### Decision · Program_Chairs · 2025-09-17

**Decision:**

Accept (poster)

**Comment:**

This paper was reviewed by four experts in the field, with 4 Boarderline Accept recommendations. All the reviewers appreciate that this paper proposes a novel approach to investigate inference time beam search for diffusion video generation, with reasonable technical improvements and relatively comprehensive experiments. Thus, this paper achieves the NeurIPS acceptance requirement. AC highly urges the authors to go through the detailed comments carefully to polish the writing and provide extra experimental details, so as to ensure the final acceptance.